# MASTERING MEMORY TASKS WITH WORLD MODELS

**Mohammad Reza Samsami**[*1,2] **Artem Zholus**[*1,3] **Janarthanan Rajendran**[1,2]
**Sarath Chandar**[1,3,4]

[1]Mila – Quebec AI Institute  [2]Université de Montréal  [3]Polytechnique Montréal  [4]CIFAR AI Chair

## ABSTRACT

Current model-based reinforcement learning (MBRL) agents struggle with long-term dependencies. This limits their ability to effectively solve tasks involving extended time gaps between actions and outcomes, or tasks demanding the recalling of distant observations to inform current actions. To improve temporal coherence, we integrate a new family of state space models (SSMs) in world models of MBRL agents to present a new method, Recall to Imagine (R2I). This integration aims to enhance both long-term memory and long-horizon credit assignment. Through a diverse set of illustrative tasks, we systematically demonstrate that R2I not only establishes a new state-of-the-art for challenging memory and credit assignment RL tasks, such as BSuite and POPGym, but also showcases superhuman performance in the complex memory domain of Memory Maze. At the same time, it upholds comparable performance in classic RL tasks, such as Atari and DMC, suggesting the generality of our method. We also show that R2I is faster than the state-of-the-art MBRL method, DreamerV3, resulting in faster wall-time convergence.

## 1 INTRODUCTION

In reinforcement learning (RL), *world models* (Kalweit & Boedecker, 2017; Ha & Schmidhuber, 2018; Hafner et al., 2019b), which capture the dynamics of the environment, have emerged as a powerful paradigm for integrating agents with the ability to perceive (Hafner et al., 2019a; 2020; 2023), simulate (Schrittwieser et al., 2020; Ye et al., 2021; Micheli et al., 2023), and plan (Schrittwieser et al., 2020) within the learned dynamics. In current model-based reinforcement learning (MBRL), the agent **learns** the world model from past experiences, enabling it to "imagine" the consequences of its actions (such as the future environment rewards and observations) and make informed decisions.

MBRL necessitates learning a world model that accurately simulates the environment's evolution and future rewards, integrating the agent's actions over long horizons. This task is compounded by the credit assignment (CA) problem, where an action's impact on future rewards must be evaluated. The agent also may need to memorize and recall past experiences to infer optimal actions. The challenge of long-term memory and CA frequently arises as a result of inadequate learning of long-range dependencies (Ni et al., 2023), due to constraints in world models' backbone network architecture.

More specifically, Recurrent Neural Networks (RNNs; Cho et al. (2014)) are employed in most MBRL methods (Ha & Schmidhuber, 2018; Hafner et al., 2019b;a; 2020; 2023) as the world models' backbone architecture because of their ability to handle sequential data. However, their efficacy is hindered by the vanishing gradients (Bengio et al., 1994; Pascanu et al., 2013). Alternately, due to the remarkable achievements of Transformers (Vaswani et al., 2017) in language modeling tasks (Brown et al., 2020; Thoppilan et al., 2022), they have been recently adopted to build world models (Chen et al., 2022; Micheli et al., 2023; Robine et al., 2023). Nonetheless, the computational complexity of Transformers is quadratic in its input sequence length. Even the optimized Transformers (Dai et al., 2019; Zaheer et al., 2021; Choromanski et al., 2022; Bulatov et al., 2022; Ding et al., 2023) become unstable during training on long sequences (Zhang et al., 2022). This prohibits Transformers-based world models from scaling to long input sequence lengths that might be required in certain RL tasks.

Recent studies have revealed that state space models (SSMs) can effectively capture dependencies in tremendously long sequences for supervised learning (SL) and self-supervised learning (SSL) tasks

---

[*]Equal contribution. {mohammad-reza.samsami, artem.zholus}@mila.quebec
See our website here: recall2imagine.github.io

(Gu et al., 2021a; Nguyen et al., 2022; Mehta et al., 2022; Smith et al., 2023; Wang et al., 2023). More specifically, the S4 model (Gu et al., 2021a) redefined the long-range sequence modeling research landscape by mastering highly difficult benchmarks (Tay et al., 2020). The S4 model is derived from a time-invariant linear dynamical system where state matrices are learned (Gu et al., 2021b). In SL and SSL tasks, it exhibits a remarkable capability to capture dependencies extending up to 16K in length, surpassing the limitations of all prior methods. Given these achievements and MBRL methods' limitations in solving memory and CA tasks, the adoption of S4 or a modified version of it is a logical decision. In this paper, we introduce a novel method termed *Recall to Imagine* (R2I), which is the first MBRL approach utilizing a variant of S4 (which was previously employed in model-free RL (David et al., 2023; Lu et al., 2024)). This method empowers agents with long-term memory. R2I emerges as a general and computationally efficient approach, demonstrating state-of-the-art (SOTA) performance in a range of memory domains. Through rigorous experiments, we demonstrate that R2I not only surpasses the best-performing baselines but also exceeds human performance in tasks requiring long-term memory or credit assignment, all while maintaining commendable performance across various other benchmarks. Our contributions can be summarized as follows:

- We introduce R2I, a memory-enhanced MBRL agent built upon DreamerV3 (Hafner et al., 2023) that uses a modification of S4 to handle temporal dependencies. R2I inherits the generality of DreamerV3, operating with fixed world model hyperparameters on every domain, while also offering an improvement in computational speed of up to 9 times.

- We demonstrate SOTA performance of the R2I agent in a diverse set of memory domains: POPGym (Morad et al., 2023), Behavior Suite (BSuite; Osband et al. (2020)), and Memory Maze (Pasukonis et al., 2022). Notably, in the Memory Maze, which is a challenging 3D domain with extremely long-term memory needed to be solved, R2I outperforms human.

- We investigate R2I's performance in established RL benchmarks, namely Atari (Bellemare et al., 2013) and DMC (Tassa et al., 2018). We show that R2I's improved memory does not compromise performance across different types of control tasks, highlighting its generality.

- We conduct ablation experiments to show the impact of the design decisions made for R2I.

## 2 BACKGROUND

### 2.1 STATE SPACE MODELS

A recent work (Gu et al., 2021a) has introduced a novel Structured State Space Sequence model (S4). This model has shown superior performance in SL and SSL tasks, compared to common deep sequence models, including RNNs, convolutional neural networks (CNNs; lec (1998)), and Transformers. It outperforms them in terms of both computational efficiency (Gu et al., 2021b) and the ability to model extremely long-range dependencies (Gu et al., 2020). S4 is a specific instance of state space models (SSMs), which can be efficiently trained by using specialized parameterization.

SSMs are derived from a linear dynamical system with control variable $u(t) \in \mathbb{R}$ and observation variable $y(t) \in \mathbb{R}$, utilizing state variables $x(t) \in \mathbb{C}^N$ for a state size $N$. The system is represented by the state matrix $\mathbf{A} \in \mathbb{C}^{N \times N}$ and other matrices $\mathbf{B} \in \mathbb{C}^{N \times 1}$, $\mathbf{C} \in \mathbb{C}^{1 \times N}$, and $\mathbf{D} \in \mathbb{R}^{1 \times 1}$:

$$x'(t) = \mathbf{A}x(t) + \mathbf{B}u(t), \quad y(t) = \mathbf{C}x(t) + \mathbf{D}u(t). \tag{1}$$

Note that these SSMs function on continuous sequences. They can be discretized by a step size $\Delta$ to allow discrete recurrent representation:

$$x_n = \bar{\mathbf{A}}x_{n-1} + \bar{\mathbf{B}}u_n, \quad y_n = \bar{\mathbf{C}}x_n + \bar{\mathbf{D}}u_n, \tag{2}$$

where $\bar{\mathbf{A}}, \bar{\mathbf{B}}, \bar{\mathbf{C}}$, and $\bar{\mathbf{D}}$ are discrete-time parameters obtained from the continuous-time parameters and $\Delta$ using methods like zero-order hold and bilinear technique (Smith et al., 2023). These representations are incorporated as a neural network layer, and each SSM is used to process a single dimension of the input sequence and map it to a single output dimension. This means that there are separate linear transformations for each input dimension, which are followed by a nonlinearity. This allows working with discrete sequence tasks, such as language modeling (Merity et al., 2016), speech classification (Warden, 2018), and pixel-level 1D image classification (Krizhevsky et al., 2009).

S4 model characterizes $\mathbf{A}$ as a matrix with a diagonal plus low-rank (DPLR) structure (Gu et al., 2021a). One benefit of this "structured" representation is that it helps preserve the sequence history;

S4 employs HiPPO framework (Gu et al., 2020) to initialize the matrix $\mathbf{A}$ with special DPLR matrices. This initialization grants the SSMs the ability to decompose $u(t)$ into a set of infinitely long basis functions, enabling the SSMs to capture long-range dependencies. Further, to make S4 more practical on modern hardware, Gu et al. (2021a) have reparameterized the mapping $u_{1:T}, x_0 \rightarrow y_{1:T}, x_T$ as a global convolution, referred to as the *convolution mode*, thereby avoiding sequential training (as in RNNs). This modification has made S4 faster to train, and as elaborated in Gu et al. (2021b), S4 models can be thought of as a fusion of CNNs, RNNs, and classical SSMs. Smith et al. (2023) uses **parallel scan** (Blelloch, 1990) to compute $u_{1:T}, x_0 \rightarrow y_{1:T}, x_{1:T}$ as efficient as **convolution mode**.

S4 has demonstrated impressive empirical results on various established SL and SSL benchmarks involving long dependencies, and it outperforms Transformers (Vaswani et al., 2017; Dao et al., 2022) in terms of inference speed and memory consumption due to its recurrent inference mode. Moreover, some recent works have focused on understanding S4 models, as well as refining them and augmenting their capabilities (Gupta et al., 2022a; Gu et al., 2022; Mehta et al., 2022; Gupta et al., 2022b; Smith et al., 2023; Ma et al., 2023). We have provided additional details in Appendix B to explain this family of S4 models. For the sake of simplicity in this study, we will be referring to all the S4 model variations as "SSMs". It is worth highlighting that a few recent methods optimize the performance of SSMs by integrating them with Transformers (Fu et al., 2023; Zuo et al., 2022; Fathi et al., 2023). This enhances the SSMs by adding a powerful local attention-based inductive bias.

## 2.2 FROM IMAGINATION TO ACTION

We frame a sequential decision-making problem as a partially observable Markov decision process (POMDP) with observations $o_t$, scalar rewards $r_t$, agent's actions $a_t$, episode continuation flag $c_t$, and discount factor $\gamma \in (0, 1)$, all following dynamics $o_t, r_t, c_t \sim p(o_t, r_t, c_t \mid o_{<t}, a_{<t})$. The goal of RL is to train a policy $\pi$ that maximizes the expected value of the discounted return $\mathbb{E}_\pi \left[ \sum_{t \geq 0} \gamma^t r_t \right]$.

In MBRL, the agent learns a model of the environment's dynamics (i.e., the world model), through an iterative process of collecting data using a policy, training the world model on the accumulated data, and optimizing the policy through the world model (Sutton, 1990; Ha & Schmidhuber, 2018). The Dreamer agent (Hafner et al., 2019a) and its subsequent versions (Hafner et al., 2020; 2023) have been impactful MBRL systems that learn the environment dynamics in a compact latent space and learn the policy entirely within that latent space. Dreamer agents consist of three primary components: the **world model**, which predicts the future outcomes of potential actions, the **critic**, which estimates the value of each state, and the **actor**, which learns to take optimal actions.

In Dreamer, an RNN-based architecture called Recurrent State-Space Model (RSSM), proposed by Hafner et al. (2019b), serves as the core of the world model, and it can be described as follows. For every time step $t$, it represents the latent state through the concatenation of deterministic state $h_t$ and stochastic state $z_t$. Here, $h_t$ is updated using a Gated Recurrent Unit (GRU; Cho et al. (2014)), and then is utilized to compute $z_t$, which incorporates information about the current observation $o_t$ and is subsequently referred to as the posterior state. Additionally, the prior state $\hat{z}_t$ which predicts $z_t$ without access to $o_t$ is computed using $h_t$. By leveraging the latent state $(z_t, h_t)$, we can reconstruct various quantities such as $o_t$, $r_t$, and $c_t$. The RSSM comprises three components: a sequence model ($h_t = f_\theta(h_{t-1}, z_{t-1}, a_{t-1})$), a representation model ($z_t \sim q_\theta(z_t \mid h_t, o_t)$), and a dynamics model ($\hat{z}_t \sim p_\theta(\hat{z}_t \mid h_t)$), where $a_{t-1}$ is the action at time step $t-1$, and $\theta$ denotes the combined parameter vector of all components. In addition to the RSSM, the world model has separate prediction heads for $o_t$, $r_t$, $c_t$. Within the *imagination* phase, it harnesses the RSSM to simulate trajectories. This is performed through an iterative computation of states $\hat{z}_t$, $h_t$ and actions $\hat{a}_t \sim \pi(\hat{a}_t \mid \hat{z}_t, h_t)$ without the need for observations (except in the initial step). The sequences of $\hat{z}_{1:T}, h_{1:T}, \hat{a}_{1:T}$ are used to train the actor and the critic. See Appendix D for more details.

## 3 METHODOLOGY

We introduce R2I (Recall to Imagine), which integrates SSMs in DreamerV3's world model, giving rise to what we term the Structured State-Space Model (S3M). The design of the S3M aims to achieve two primary objectives: capturing long-range relations in trajectories and ensuring fast computational performance in MBRL. S3M achieves the desired speed through parallel computation during training and recurrent mode in inference time, which enables quick generation of imagined trajectories. In Figure 1, a visual representation of R2I is provided, and we will now proceed to describe its design.

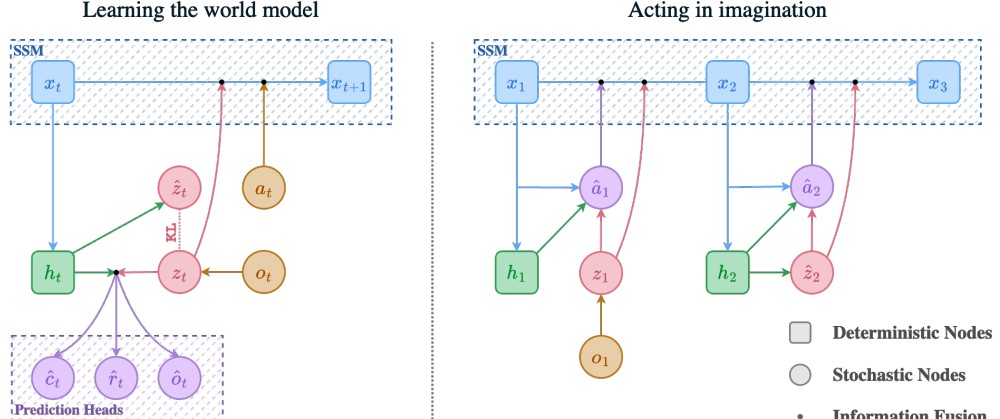

Figure 1: Graphical representation of R2I. **(Left)** The world model encodes past experiences, transforming observations and actions into compact latent states. Reconstructing the trajectories serves as a learning signal for shaping these latent states. **(Right)** The policy learns from trajectories based on latent states imagined by the world model. The representation corresponds to the full state policy, and we have omitted the critic for the sake of simplifying the illustration.

## 3.1 WORLD MODEL DETAILS

**Non-recurrent representation model.** Our objective when updating the world model is to calculate S3M deterministic states $h_{1:T}$ in parallel by simultaneously feeding all actions $a_t$ and stochastic state $z_t$, where $T$ represents the length of the entire sequence. We aim to carry out this computation as $h_{1:T}, x_{1:T} = f_\theta((a_{1:T}, z_{1:T}), x_0)$ where $x_t$ is a hidden state and $f_\theta$ is a sequence model with a SSM network. To achieve this, prior access to all actions $a_{1:T}$ and stochastic states $z_{1:T}$ is required. However, we encounter a challenge due to the sequential nature of the relationship between the representation model $q_\theta(z_t \mid h_t, o_t)$ and sequence model $f_\theta(h_{t-1}, z_{t-1}, a_{t-1})$: at time step $t$, the representation model's most recent output, denoted as $z_{t-1}$, is fed into the sequence model, and the resulting output $h_t$ is then used within the representation model to generate $z_t$. Hence, similar to Chen et al. (2022); Micheli et al. (2023); Robine et al. (2023); Deng et al. (2023), by eliminating the dependency on $h_t$ in the representation model, we transform it to a non-recurrent representation model $q_\theta(z_t \mid o_t)$. This modification allows us to compute the posterior samples independently for each time step, enabling simultaneous computation for all time steps. By utilizing a parallelizable function $f_\theta$, we can then obtain $h_{1:T}$ in parallel. Appendix M includes a systematic analysis to investigate how this modification impacts the performance of the DreamerV3 across a diverse set of tasks. The results indicate that transforming $q_\theta(z_t \mid o_t, h_t)$ to $q_\theta(z_t \mid o_t)$ does not hurt the performance.

**Architecture details.** Inspired by Dreamer, R2I's world model consists of a representation model, a dynamics model, and a sequence model (together forming S3M). In addition to that, there are three prediction heads: an observation predictor $p_\theta(\hat{o}_t \mid z_t, h_t)$, a reward predictor $p_\theta(\hat{r}_t \mid z_t, h_t)$, and an episode continuation predictor $p_\theta(\hat{c}_t \mid z_t, h_t)$. At each time step, S3M processes a pair of $(a_t, z_t)$ to output the deterministic state $h_t$. Inside, it operates over the hidden state $x_t$, so it can be defined as $h_t, x_t = f_\theta((a_{t-1}, z_{t-1}), x_{t-1})$. Specifically, $f_\theta$ is composed of multiple layers of SSMs, each one calculating outputs according to Equation 2. The outputs are then passed to GeLU (Hendrycks & Gimpel, 2023), which is followed by a fully-connected GLU transformation (Dauphin et al., 2017), and finally by a LayerNorm (Ba et al., 2016). This follows the architecture outlined by Smith et al. (2023). The deterministic state $h_t$ is the output from the final SSM layer. The set of all SSM layer hidden states is denoted $x_t$. See Appendix B.1 for SSMs design details. In image-based environments, we leverage a CNN encoder for $q_\theta(z_t \mid o_t)$ and a CNN decoder for $p_\theta(\hat{o}_t \mid z_t, h_t)$. In contrast, in tabular environments, both $q_\theta(z_t \mid o_t)$ and $p_\theta(\hat{o}_t \mid z_t, h_t)$ are MLPs. We include the details on network widths, depths, and other hyperparameters in Appendix E.

**Training details.** R2I optimizes the following objective:

$$\mathcal{L}(\theta) = \mathbb{E}_{z_{1:T} \sim q_\theta} \sum_{t=1}^{T} \mathcal{L}^{\text{pred}}(\theta, h_t, o_t, r_t, c_t, z_t) + \mathcal{L}^{\text{rep}}(\theta, h_t, o_t) + \mathcal{L}^{\text{dyn}}(\theta, h_t, o_t) \tag{3}$$

$$\mathcal{L}^{\text{pred}}(\theta, h_t, o_t, r_t, c_t, z_t) = -\beta_{\text{pred}}(\ln p_\theta(o_t \mid z_t, h_t) + \ln p_\theta(r_t \mid z_t, h_t) + \ln p_\theta(c_t \mid z_t, h_t)) \quad (4)$$

$$\mathcal{L}^{\text{dyn}}(\theta, h_t, o_t) = \beta_{\text{dyn}} \max(1, \text{KL}[\text{sg}(q_\theta(z_t \mid o_t)) \,\|\, \quad p(z_t \mid h_t) \;]) \quad (5)$$

$$\mathcal{L}^{\text{rep}}(\theta, h_t, o_t) = \beta_{\text{rep}} \max(1, \text{KL}[\quad q_\theta(z_t \mid o_t) \,\|\, \text{sg}(p(z_t \mid h_t))]) \quad (6)$$

$$h_{1:T}, x_{1:T} = f_\theta((a_{1:T}, z_{1:T}), x_0) \quad (7)$$

Here, sg represents the stop gradient operation. This loss, resembling the objective utilized in (Hafner et al., 2023), is derived from Evidence Lower Bound (ELBO), but our objective differs from ELBO in three ways. First, we clip KL-divergence when it falls below the threshold of 1(Hafner et al., 2020; 2023). Secondly, we use KL-balancing (Hafner et al., 2020; 2023) to prioritize the training of the S3M. Third, we use scaling coefficients $\beta_{\text{pred}}, \beta_{\text{rep}}, \beta_{\text{dyn}}$ to adjust the influence of each term in the loss function (Higgins et al., 2017; Hafner et al., 2023). Some works on SSMs recommend optimizing state matrices using a smaller learning rate; however, our experiments indicate that the most effective approach is to use the same learning rate used in the rest of the world model.

**SSMs Computational Modeling.** To enable the parallelizability of world model learning, as outlined in Section 2.1, we have the option to select between two distinct approaches: convolution (Gu et al., 2021a) and parallel scan (Smith et al., 2023). After thorough deliberation, we opted for parallel scan due to several compelling reasons. Firstly, as we discuss later in Section 3.2, *it is essential to pass hidden states $x_t$ to the policy in memory environments*, a critical finding we empirically analyze in Appendix N. Another consequence of not yielding $x_t$ via convolution mode is that it would necessitate several burn-in steps to obtain correct hidden states, akin to Kapturowski et al. (2019), resulting in quadratic imagination complexity. Furthermore, parallel scan enables scaling of sequence length in batch across distributed devices, a capability not supported by the convolution mode. Table 6 summarizes computational complexities associated with different types of recurrences, including RNNs, SSMs, and Attention used in studies like Chen et al. (2022).

| Method | Training | Inference step | Imagination step | Parallel | State Reset |
|---|---|---|---|---|---|
| Attn | $\mathcal{O}(L^2)$ | $\mathcal{O}(L^2)$ | $\mathcal{O}((L+H)^2)$ | ✓ | ✓ |
| RNN | $\mathcal{O}(L)$ | $\mathcal{O}(1)$ | $\mathcal{O}(1)$ | ✗ | ✓ |
| SSM (Conv) | $\mathcal{O}(L)$ | $\mathcal{O}(1)$ | $\mathcal{O}(L)$ | ✓ | ✗ |
| SSM (Par.Scan) | $\mathcal{O}(L)$ | $\mathcal{O}(1)$ | $\mathcal{O}(1)$ | ✓ | ✓ |

Table 1: The asymptotic runtimes of different architectures. $L$ is the sequence length and $H$ is the imagination horizon. The outer loop of the imagination process cannot be parallelized. Attention and SSM+Conv accept the full context of $\mathcal{O}(L + H)$ burn-in and imagined steps which results in $\mathcal{O}((L + H)^2)$ step complexity for Attention and $\mathcal{O}(L)$ for SSM+Conv. SSMs combine compact recurrence with parallel computation reaching the best asymptotical complexity.

Finally, parallel scan can facilitate the resetting of hidden states. When sampling a sequence from the buffer, it may comprise of multiple episodes; thus, the hidden states coming from terminal states to the initial states in new episodes must be reset. This boosts the early training performance, when the episodes may be short. Inspired by Lu et al. (2024), we modify the SSM inference operator to support resetting hidden states. Achieving this is not feasible with convolution mode. Details of our SSMs operator used by the parallel scan is provided in Appendix C.

## 3.2 ACTOR-CRITIC DETAILS

In the design of Dreamer's world model, it is assumed that $h_t$ contains information summarizing past observations, actions, and rewards. Then, $h_t$ is leveraged in conjunction with the stochastic state $z_t$ to reconstruct or predict observations, rewards, episode continuation, actions, and values. Unlike DreamerV3, which utilizes a GRU cell wherein $h_t$ is passed both to the reconstruction heads and the next recurrent step, R2I exclusively passes $h_t$ to prediction heads, while SSM's hidden state $x_t$ is used in the next recurrent update of S3M. This implies that the information stored in $h_t$ and $x_t$ could potentially vary. Empirically, we discovered that this difference can lead to the breakdown of policy learning when using $\pi(\hat{a}_t \mid z_t, h_t)$, but it remains intact when we use $\pi(\hat{a}_t \mid z_t, x_t)$ in memory-intensive environments. Surprisingly, we found that incorporating all features into the policy $\pi(\hat{a}_t \mid z_t, h_t, x_t)$ is not a remedy. The reason lies in the non-stationarity of these features; their empirical distribution changes over time as the world model trains, ultimately leading to instability in the policy training process. A similar phenomenon was also observed in Robine et al. (2023). We study the dependency of policy features on the performance in Appendix N, where we cover a diverse set of environments: from non-memory vector-based ones to image-based memory environments. In different environments, we condition the policy and value function on the information from S3M

in the following ways: we use the *output state policy* that takes $(z_t, h_t)$ as input, the *hidden state policy* that takes $(z_t, x_t)$ as input, and the *full state policy* that takes $(z_t, h_t, x_t)$ as input. To train actor-critic, we opt for the procedure proposed in DreamerV3 (Hafner et al., 2023). For a detailed description of the actor-critic training process, refer to Appendix D.

## 4 EXPERIMENTS

We conduct a comprehensive empirical study to assess the generality and memory capacity of R2I across a wide range of domains, including credit assignment, memory-intensive tasks, and non-memory tasks, all while maintaining fixed hyperparameters of the world model. We cover five RL domains: BSuite (Osband et al., 2020), POPGym (Morad et al., 2023), Atari 100K (Łukasz Kaiser et al., 2020), DMC Tassa et al. (2018), and Memory Maze (Pasukonis et al., 2022). The section is organized as follows. In Sections 4.1 and 4.2, we evaluate R2I's performance in two distinct memory-intensive settings: simple tabular environments and complex 3D environments. We show that not only does R2I achieve the SOTA performance, but it also surpasses human-level performance in the complex Memory Maze domain. In Section 4.3, we demonstrate that we do not trade the generality for improved memory capabilities. **Figure 2 shows R2I's impressive computational efficiency, with a speed increase of up to 9 times compared to its predecessor, DreamerV3**. Note that the image environments are representative of Memory Maze, and the vector environments represent POPGym.

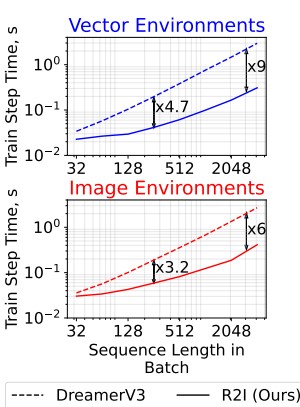

Figure 2: Computational time taken by DreamerV3 and R2I (lower is preferred)

We reuse most of the world model hyperparameters from DreamerV3. In all environments, we use a First-in First-out (FIFO) replay buffer size of 10M steps to train R2I. We found this helps stabilize the world model and prevent overfitting on a small buffer. Also, we vary features that the policy is conditioned on (i.e., output state policy $\pi(\hat{a}_t \mid z_t, h_t)$, hidden state policy $\pi(\hat{a}_t \mid z_t, x_t)$, or full state policy $\pi(\hat{a}_t \mid z_t, h_t, x_t)$). Our primary takeaway is to leverage the output state policy in non-memory environments and the full state policy or hidden state policy within memory environments, as explained in Section 3.2. We also found that even in memory environments, the full state policy cannot be preferred over the hidden state policy because of the instability of features – since the world model is trained alongside the policy, the former might change the feature distribution which introduces non-stationarity for the policy.

### 4.1 QUANTIFYING MEMORY OF R2I

In this section, we study the performance of R2I in challenging memory environments of BSuite and POPGym domains, which are tabular environments. Despite their simplicity, these environments pose a challenge for MBRL algorithms since the world model needs to learn causal connections over time. While SSMs have shown their ability to handle extremely long-range dependencies in SL and SSL (Gu et al., 2021a), this capability does not necessarily translate to MBRL, even though the world model optimizes the same supervised objective. This discrepancy arises from the lifelong nature of world model training. That is, it needs to boot-

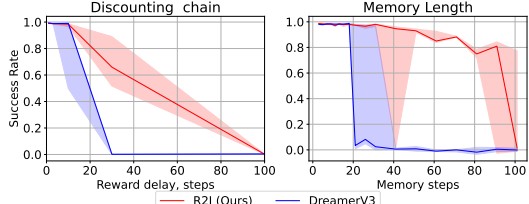

Figure 3: Success rates of DreamerV3 (which holds the previous SOTA) and R2I in BSuite environments. A separate model is trained for every point on the x-axis. A median value (over 10 seeds) is plotted filling between 25-th and 75-th percentiles. Training curves are in Appendix F.

strap its performance from a very small dataset with hugely imbalanced reward "labels" (as opposed to big and well-balanced long-range datasets on which SSMs shine (Tay et al., 2021)). Additionally, the continuously growing replay buffer imposes the need to quickly learn the newly arrived data which requires an ability for quick adaptation of the world model throughout its optimization. The section's goal is to give an insight into how extensive are R2I's memory capabilities.

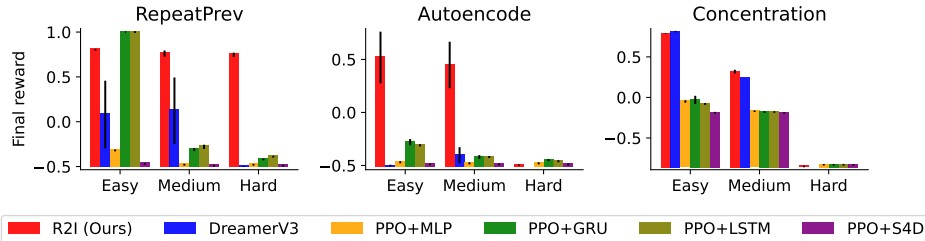

Figure 4: R2I results in memory-intensive environments of POPGym. Our method establishes the new SOTA in the hardest memory environments; `Autoencode`: `-Easy`, `-Medium`; `RepeatPrevious`: `-Medium`, `-Hard`; `Concentration`: `-Medium`. Note that `Concentration` is a task that can be partially solved without memory. For `PPO+S4D`, refer to Appendix S.

**Behavior Suite experiments.** To study the ability of the R2I model to handle longer episodes, we conduct quantitative experiments within a subset of the BSuite environments. These environments are specifically designed to evaluate an agent's memory capacity and its ability to effectively perform credit assignment. In particular, we carry out experiments within `Memory Length` and `Discounting Chain` environments. The former focuses on memory, and the latter serves as a credit assignment task. In `Memory Length` environment, the goal is to output an action which is dictated by the initial observation (the episode length i.e., the *memory steps number* is an environment parameter). Essentially, the agent must carry the information from the initial observation throughout the entire episode. In the `Discounting Chain`, the first action (which is categorical) causes a reward that is only provided after a certain number of steps, specified by the parameter *reward delay*.

As depicted in Figure 3, the previous SOTA DreamerV3 learns the dependencies between actions and rewards in both `Discounting Chain` and `Memory Length` with reward delays of up to 30 environment steps. Note that every run either converged to a maximum reward or failed (based on the random seed). We plot the success rate as the fraction of runs that achieved success. R2I excels in both tasks, significantly outperforming in the preservation of its learning ability across a wider range of varying environment complexities. In these experiments, we leverage the output state policy (i.e., operating on latent variable $z_t$ and S3M output $h_t$). More details are provided in Appendix E.

**POPGym experiments.** We perform a study to assess R2I in a more challenging benchmark, namely, POPGym (Morad et al., 2023). This suite offers a range of RL environments designed to assess various challenges related to POMDPs, such as navigation, noise robustness, and memory. Based on Ni et al. (2023), we select the three most memory-intensive environments: `RepeatPrevious`, `Autoencode`, and `Concentration`. These environments require an optimal policy to memorize the highest number of events (i.e., actions or observations) at each time step. Each environment in POPGym has three difficulty levels: `Easy`, `Medium`, and `Hard`. In the memory environments of this study, the complexity is increased by the number of actions or observations that the agent should keep track of *simultaneously*. All environments in this study have categorical observation and action spaces. A detailed explanation of the environments is provided in Appendix G.

As POPGym was not included in the DreamerV3 benchmark, we performed hyperparameter tuning of both DreamerV3 and R2I, solely on adjusting the network sizes of both. This is because DreamerV3 is a generalist agent that works with a fixed set of hyperparameters and in this environment, with sizes primarily influencing its data efficiency. We observed a similar characteristic in R2I. The results of hyperparameter tuning are available in Appendix L. For R2I, we use the hidden state policy: $\pi(\hat{a}_t \mid z_t, h_t)$ as we found it much more performant, especially in memory-intensive tasks (see Appendix N for policy inputs ablations). We train R2I in POPGym environments using a unified and fixed set of hyperparameters. In addition to R2I and DreamerV3, we include model-free baselines from Morad et al. (2023). These include PPO (Schulman et al., 2017) model-free policy with different observation backbones, such as GRU, LSTM, MLP, and MLP with timestep number added as a feature (PosMLP). PPO with GRU is the best-performing model-free baseline of POPGym while PPO+LSTM is the second best. PPO+MLP and PPO+PosMLP are included for a sanity check - the better their performance is, the less is the memory needed in the environment.

---

[1]A policy without any memory exists that outperforms a random policy but underperforms the optimal one.

As illustrated in Figure 4, R2I demonstrates the new SOTA performance, outperforming every baseline in `Autoencode`, `Easy` and `Medium` tasks. Note that R2I outperforms all 13 model-free baselines of the POPGym benchmark by a huge margin (we did not include them due to space constraints). R2I also shows consistently strong performance in `RepeatPrevious` tasks, setting a new SOTA in both `Medium` and `Hard` (compared to all 13 model-free baselines and DreamerV3). In `Concentration`, the model-free memory baselines fail to outperform a simple MLP policy, suggesting that they all converge to a non-memory-based suboptimal policy. R2I advances this towards a better memory policy. Its performance is roughly equal to DreamerV3 in an `Easy` and slightly better in the `Medium` task. As Appendix G suggests, all `RepeatPrevious` tasks require up to 64 memorization steps, while `Autoencode Easy` and `Medium` require up to 104. In `Concentration Easy` and `Medium` this length is up to 208 steps, however, since PPO+MLP shows somewhat good performance, likely less than 208 memorization steps are required. This observation is consistent with the results of the BSuite experiments, which demonstrate that our model is capable of memorizing up to approximately 100 steps in time. To summarize, **these results indicate that R2I significantly pushes the memory limits**.

## 4.2 EVALUATING LONG-TERM MEMORY IN COMPLEX 3D TASKS

Memory Maze (Pasukonis et al., 2022) presents randomized 3D mazes where the egocentric agent is repeatedly tasked to navigate to one of multiple objects. For optimal speed and efficiency, the agent must retain information about the locations of objects, the maze's wall layout, and its own position. Each episode can extend for up to 4K environment steps. An ideal agent equipped with long-term memory only needs to explore each maze once, a task achievable in a shorter time than the episode's duration; subsequently, it can efficiently find the shortest path to reach each requested target. This task poses a fundamental challenge for existing memory-augmented RL algorithms, which fall significantly behind human performance in these tasks.

In this benchmark, we found that DreamerV3 works equally well as DreamerV2 reported in Pasukonis et al. (2022). Therefore, we use the size configuration of Dreamer outlined in Pasukonis et al. (2022). Note that this baseline also leverages truncated backpropagation through time (TBTT), a technique demonstrated to enhance the preservation of information over time (Pasukonis et al., 2022). We use the "medium memory" size configuration of R2I in this work (see Table 2 in Appendix). We use the full state policy ($\pi(\hat{a}_t \mid z_t, h_t, x_t)$ i.e., conditioning on stochastic state, and S3M output, and hidden states at step $t$) in this environment. We trained and tested R2I and other methods on 4 existing maze sizes: 9x9, 11x11, 13x13, and 15x15. The difference between them is in the number of object rooms and the episode lengths. More difficult maze sizes have more environment steps in the episode making it more challenging to execute a successful series of object searches. R2I and other baselines are evaluated after 400M environment steps or two weeks of training. We also compare R2I with IMPALA (Espeholt et al., 2018), which is the leading model-free approach (Pasukonis et al., 2022).

As shown in Figure 5, R2I consistently outperforms baseline methods in all of these environments. In 9x9 mazes, it demonstrates performance similar to the Dreamer, while significantly outperforming IMPALA. In 11x11, 13x13, and 15x15 mazes, it has a remarkably better performance than both baselines. Moreover, it has surpassed human-level abilities in solving 9x9, 11x11, and 13x13 mazes. **These results establish R2I as a SOTA in this complex 3D domain**.

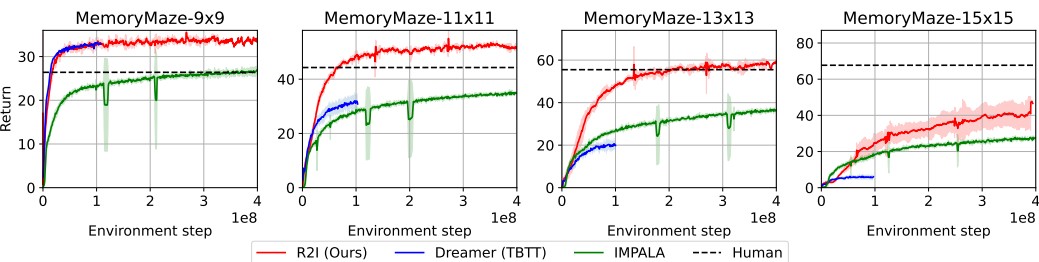

Figure 5: Scores in Memory Maze after 400M environment steps. R2I outperforms baselines across difficulty levels, becoming the domain's new SOTA. Due to its enhanced computational efficiency, R2I was trained during a fewer number of days compared to Dreamer, as illustrated in Figure 26.

### 4.3 ASSESSING THE GENERALITY OF R2I IN NON-MEMORY DOMAINS

We conduct a sanity check by assessing R2I's performance on two widely used RL benchmarks: Atari (Bellemare et al., 2013) and DMC (Tassa et al., 2018), as parts of the DreamerV3 benchmark (Hafner et al., 2023). Even though these tasks are nearly fully observable and do not necessitate extensive memory to solve (it is often enough to model the dynamics of only the last few steps), evaluating R2I on them is essential as we aim to ensure our agent's performance across a wide range of tasks that require different types of control: continuous control (in DMC) and discrete (in Atari).

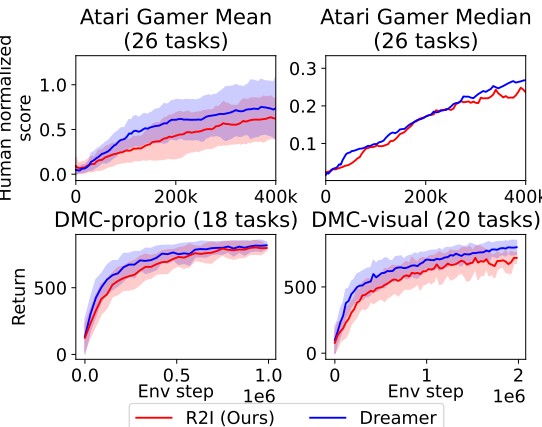

In all the experiments conducted within Atari 100K (Łukasz Kaiser et al., 2020) and DMC, we fix hyperparameters of the world model. In Atari and the proprio benchmark in DMC, we utilize output state policies, as we found them

Figure 6: Average performance in Atari and DMC. For full training curves, see Appendix K (Atari), Appendix I (DMC-P), and Appendix J (DMC-V)

more performant (for ablations with different policy types, see Appendix N). In the visual benchmark in DMC, we use hidden state policy. Note that for continuous control, the policy is trained via differentiating through the learned dynamics. R2I maintains a performance similar to DreamerV3 in these domains, as demonstrated in Figure 6, implying that in the majority of standard RL tasks (see Appendix Q), **R2I does not sacrifice generality for improved memory capabilities**.

## 5 CONCLUSION

In this paper, we introduced R2I, a general and fast model-based approach to reinforcement learning that demonstrates superior memory capabilities. R2I integrates two strong algorithms: DreamerV3, a general-purpose MBRL algorithm, and SSMs, a family of novel parallelizable sequence models adept at handling extremely long-range dependencies. This integration helps rapid long-term memory and long-horizon credit assignment, allowing R2I to excel across a diverse set of domains, all while maintaining fixed hyperparameters across all domains. Through a systematic examination, we have demonstrated that R2I sets a new state-of-the-art in domains demanding long-term temporal reasoning: it outperforms all known baselines by a large margin on the most challenging memory and credit assignment tasks across different types of memory (long-term and short-term) and observational complexities (tabular and complex 3D). Remarkably, it transcends human performance in complex 3D tasks. Furthermore, we have demonstrated that R2I achieves computation speeds up to 9 times faster than DreamerV3.

Our study presents the first model-based RL approach that uses SSMs. While R2I offers benefits for improving memory in RL, it also has limitations, which we leave for future research. For instance, it can be explored how R2I can be augmented with attention mechanisms, given that Transformers and SSMs exhibit complementary strengths (Mehta et al., 2022). As mentioned in Section 2.1, hybrid architectures have been introduced in language modeling tasks. Moreover, the sequence length within the training batches for world model learning is not currently extremely long, as is the horizon (i.e., the number of steps) of imagination in actor-critic learning. Future work could focus on these aspects to further enhance memory capabilities.

## ACKNOWLEDGEMENTS

We thank Albert Gu for his thorough and insightful feedback on the SSM part of the project. We also acknowledge The Annotated S4 Blog (Rush & Karamcheti, 2022) and S5 codebase (Smith et al., 2023) which inspired our JAX implementation. We also thank Danijar Hafner, Steven Morad, Ali Rahimi-Kalahroudi, Michel Ma, Tianwei Ni, Darshan Patil, and Roger Creus for their helpful feedback on our method and the early draft of the paper. We thank Jurgis Pasukonis for sharing the data for memory maze baseline plots. This research was enabled by computing resources provided by Mila (mila.quebec), the Digital Research Alliance of Canada (alliancecan.ca), and NVIDIA (nvidia.com). We thank Mila's IDT team, and especially Olexa Bilaniuk for helping with numerous technical questions during this work and especially for the help in the implementation of the new I/O efficient RL replay buffer. Janarthanan Rajendran acknowledges the support of the IVADO postdoctoral fellowship. Sarath Chandar is supported by the Canada CIFAR AI Chairs program, the Canada Research Chair in Lifelong Machine Learning, and the NSERC Discovery Grant.

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

# Appendix

## Table of Contents

## A    RELATED WORK

Addressing sequential decision-making problems requires the incorporation of temporal reasoning. This necessitates a crucial element of intelligence - *memory* (Parisotto et al., 2020; Esslinger et al., 2022; Morad et al., 2023). Essentially, memory enables humans and machines to retain valuable information from past events, empowering us to make informed decisions at present. But it does not stop there; another vital aspect is the ability to gauge the effects and consequences of our past actions and choices on the feedback/outcome we receive now - be it success or failure. This process is termed *credit assignment* (Hung et al., 2019; Arjona-Medina et al., 2019; Widrich et al., 2021; Raposo et al., 2021). In their recent work, Ni et al. (2023) explore the relationship between these two concepts, examining their interplay. The process of learning which parts of history to remember is deeply connected to the prediction of future rewards. Simultaneously, mastering the skill of distributing current rewards to past actions inherently involves a memory component.

To improve temporal reasoning, we leverage model-based RL (Moerland et al., 2022). The essence of MBRL lies in its utilization of a world model (Kalweit & Boedecker, 2017; Ha & Schmidhuber, 2018; Hafner et al., 2019b). The world model is employed to plan a sequence of actions that maximize the task reward (Hafner et al., 2019a; 2020; 2023; Schrittwieser et al., 2020; Ye et al., 2021; Micheli et al., 2023). Our focus on MBRL stems from its potential in sample efficiency, reusability, transferability, and safer planning, along with the advantage of more controllable aspects due to its explicit supervised learning component (i.e., world modeling). Within the realm of MBRL methodologies, we decided to build upon the SOTA DreamerV3 framework (Hafner et al., 2023), which has proven capacities for generalization, sample efficiency, and scalability.

MBRL researchers commonly utilize RNNs (Ha & Schmidhuber, 2018; Hafner et al., 2019a; 2020; 2023) or Transformers (Chen et al., 2022; Micheli et al., 2023; Robine et al., 2023) as the backbone of their world models to integrate temporal reasoning into the agent. However, both of these approaches face limitations when it comes to modeling long-range dependencies, as stated earlier in Section 1. Recent research indicates that SSMs (Gu et al., 2021a;b; Gupta et al., 2022a; Gu et al., 2022; Mehta et al., 2022; Gupta et al., 2022b; Smith et al., 2023; Ma et al., 2023) can replace Transformers, capturing dependencies in very long sequences more efficiently (sub-quadratic complexity) and in parallel. Thus, they can be a good candidate for a backbone architecture of world models. Concurrently with this work, S4WM (Deng et al., 2023) has also explored the utilization of SSMs for enhancing world models within the context of SSL, a subtask in MBRL. While S4WM is an improvement in the task of world modeling, it does not extend to proposing methods for MBRL. Furthermore, enhanced world modeling alone does not inherently ensure increased performance in RL tasks (Nikishin et al., 2021), a point that is further elaborated upon in Appendix P. This suggests that despite the progress in world modeling accuracy and efficiency, additional strategies must be integrated to translate these improvements into tangible RL gains, as detailed in Appendix P.1.

## B    VARIATIONS OF SSMS AND OUR DESIGN CHOICES

Structured State Space Sequence model (S4; Gu et al. (2021a)) is built upon classical state space models, which are commonly used in control theory (Brogan, 1991). As mentioned in the Section 2.1, S4 is built upon three pivotal and independent pillars:

- **Parametrization of the structure**. A pivotal aspect is how we parameterize the matrices. S4 conjugates matrices into a diagonal plus low-rank (DPLR).

- **Dimensionality of the SSM**. S4 utilizes separate linear transformations for each input dimension, known as single-input single-output (SISO) SSMs.

- **Computational modeling**. The third axis revolves around how we model the computation process to parallelize. Gu et al. (2021a) model the computation as a global convolution.

Recent works have introduced certain modifications to S4, specifically targeting these pillars. Some research (Gupta et al., 2022a; Smith et al., 2023) demonstrate that employing a diagonal approximation for the HiPPO (Gu et al., 2020) yields comparable performance and thus represents the state matrices as diagonal, instead of DPLR.

Regarding the second axis, Smith et al. (2023) tensorize the 1-D operations into a multi-input multi-output (MIMO) SSM, facilitating a more direct and simplified implementation of SSMs. Furthermore, Smith et al. (2023) replace the global convolution operation with a parallel scan operation, thereby eliminating the complex computation involved in the convolution kernel.

The synergy between these axes of variations may significantly influence the agent's behavior and performance. In the following section, we will demonstrate our process for choosing each option and examine how each axis influences the performance. We further will provide more in-depth information regarding our choices for SSM hyperparameters.

### B.1 DESIGN DECISIONS FOR SSM

#### B.1.1 SSM PARAMETRIZATION

While parameterizing the state matrix as a DPLR matrix could potentially offer a more expressive representation compared to a diagonal SSM, our decision was to diagonalize it. This decision was made to simplify control while maintaining high performance. Notably, the works of Gupta et al. (2022a) and Gu et al. (2022) demonstrate the feasibility of achieving S4's level of performance by employing a significantly simpler, fully diagonal parameterization for state spaces.

#### B.1.2 SSM DIMENSIONALITY

We decided to employ MIMO SSMs, a choice influenced by a convergence of factors. Firstly, by utilizing MIMO, the latent size can be substantially reduced, optimizing computational resources and accelerating processing. As a result, the implementation of efficient parallelization is facilitated.

Further, MIMO SSMs eliminate the need for mixing layers and offer the flexibility to accommodate diverse dynamics and couplings of input features within each layer. This stands in contrast to independent SISO SSMs, which process each channel of input features separately and then combine them using a mixing layer. Accordingly, the MIMO approach enables distinct processing tailored to individual input channels, leading to enhanced model adaptability and improved feature representation (Smith et al., 2023).

To validate the benefits of this decision, an ablation study was conducted. This comparative analysis showcases the differences between MIMO and SISO SSMs, empirically demonstrating the advantages of the MIMO.

## C MANAGING SEVERAL EPISODES IN A SAMPLED SEQUENCE

As mentioned in Section 3, handling episode boundaries in the sampled sequences necessitates the world model's capability to reset the hidden state. Given our utilization of parallel scan (Blelloch, 1990) for SSM computational modeling, it follows that adaptations to the binary operator in Smith et al. (2023) are essential. As a reminder, SSMs unroll a linear time-invariant dynamical system in the following form:

$$x_n = \bar{\mathbf{A}} x_{n-1} + \bar{\mathbf{B}} u_n, \quad y_n = \bar{\mathbf{C}} x_n + \bar{\mathbf{D}} u_n$$

Smith et al. (2023) employ parallel scans for the efficient computation of SSM states, denoted as $x_n$. Parallel scans leverage the property that "associative" operations can be computed in arbitrary order. When employing an associative binary operator denoted as $\bullet$ on a sequence of $L$ elements $[e_1, e_2, \ldots, e_L]$, parallel scan yields $[e_1, (e_1 \bullet e_2(, \ldots, (e_1 \bullet e_2 \bullet \cdots \bullet e_L))]$. It is worth mentioning that, parallel scan can be computed with a complexity of $\mathcal{O}(\log L)$. In Smith et al. (2023), the elements $e_i$ and the operator $\bullet$ are defined as follows:

$$e_n = (e_{n,0}, e_{n,1}) = (\bar{A}, \bar{\mathbf{B}} u_n), \quad e_i \bullet e_j = \left( e_{j,0} \times e_{i,0}, e_{j,0} \cdot e_{i,1} + e_{j,1} \right)$$

Here, $\times$ is the matrix-matrix product, and $\cdot$ is the matrix-vector product. To make resettable states, we need to incorporate the "done" flag while ensuring the preservation of the associative property. To do so, every element $e_n$ is defined as follows:

$$e_n = (e_{n,0}, e_{n,1}, e_{n,2}) = (\bar{\mathbf{A}}, \bar{\mathbf{B}} u_n, d_n)$$

where $d_n$ represents the done flag for the $n$th element. Now, we present the modified binary operator •, defined as:

$$e_i \bullet e_j := \big((1 - d_i) \cdot e_{j,0} \times e_{i,0} + e_{i,2} \cdot e_{j,0},$$
$$(1 - d_i)e_{j,0} \cdot e_{i,1} + e_{j,1},$$
$$\texttt{clip}(e_{i,2} + e_{j,2}, 0, 1)\big)$$

Where the clip function clips the value between zero and one. It is important to highlight this reparameterization of the operator defined in Lu et al. (2024), eliminates the need for if/else conditions. Thus, as discussed in Lu et al. (2024), the modified operator remains associative and retains the desirable properties we aimed to achieve.

Unlike Lu et al. (2024), we no longer assume a hidden state initialized to $x_0 = 0$; hence, a different initialization becomes necessary, specifically $e_0 = (\mathbf{I}, x_0, 0)$.

## D  TRAINING ACTOR-CRITIC

The policy and value functions are trained using latent imagination, regardless of their input features. This process aligns with the methodology outlined in Section 2.2. It begins with the generation of initial stochastic states $z_{1:T} \sim q_\theta(z_{1:T} \mid o_{1:T})$ and the computation of the sequence of deterministic hidden states $h_{1:T}, x_{1:T} = f_\phi((a_{1:T}, z_{1:T}), x_0)$ through a parallel scan operation. We reuse the previously computed values of $z_{1:T}$, $h_{1:T}$, and $x_{1:T}$ during training, which the parallel scan facilitates, thus eliminating the requirement for several burn-in steps before imagination.

Denoting $x_{j|t}$ as the state of $x$ after $j$ steps of imagination, given $t$ context steps, we initialize with $x_{0|t} = x_t$, $h_{0|t} = h_t$, and $z_{0|t} = z_t$. We then compute the action $\hat{a}_{0|t} \sim \pi(\hat{a} \mid z_{0|t}, h_{0|t}, x_{0|t})$ and after $h_{1|t}, x_{1|t} = f_\phi((\hat{a}_{0|t}, z_{0|t}), x_{0|t}); \hat{z}_{1|t} \sim p_\theta(\hat{z} \mid h_{1|t})$. This process continues for $H$ steps, during which we compute $\hat{a}_{1:H|t}, \hat{z}_{1:H|t}, h_{1:H|t}$, rewards and continue flags are computed via $\hat{r}_{1:H|t} \sim p(\hat{r} \mid \hat{z}_{1:H|t}, h_{1:H|t})$, $\hat{c}_{1:H|t} \sim p(\hat{c} \mid \hat{z}_{1:H|t}, h_{1:H|t})$, respectively. The ultimate goal is to train the policy to maximize the estimated return that follows:

$$R^\lambda_{j|t} = \hat{r}_{j|t} + c_{j|t}\left((1 - \lambda)v(\hat{z}_{j|t}, h_{j|t}, x_{j|t}) + \lambda R^\lambda_{j+1|t}\right), \quad R^\lambda_{H|t} = v(\hat{z}_{j|t}, h_{j|t}, x_{T|t}) \quad (8)$$

To train the actor and critic, we employ Reinforce (Williams, 1992) for scenarios with discrete action spaces, coupled with the method of backpropagation through latent dynamics as introduced by Hafner et al. (2020). Our approach adheres to the DreamerV3 protocol of training the actor and critic networks (Hafner et al., 2023), which includes utilization of the fixed entropy bonus, the use of twohot regression in critic, and the employment of percentile-based normalization for the returns.

# E  ALGORITHM HYPERPARAMETERS

|  | Small Memory | Small | Medium Memory |
|---|---|---|---|
| Used in | BSuite and POPgym | Atari and DMC | Memory Maze |
| $h_t$ size | 512 | 512 | 2048 |
| $x_t$ size, per layer | 512 | 192 | 512 |
| SSM layers | 3 | 5 | 5 |
| SSM units | 1024 | 512 | 1024 |

Table 2: S3M size configurations employed in this study.

| Name | Value |
|---|---|
| FIFO replay buffer size | $10^7$ |
| Batch length, $L$ | 1024 |
| Batch size | 4 |
| Nonlinearity | LayerNorm + SiLU |
| SSM discretization method | bilinear |
| SSM nonlinearity | GeLU + GLU + LayerNorm |
| SSM matrices parameterization | Diagonal |
| SSM dimensionality parameterization | MIMO |
| SSM matrix blocks number (HiPPOs number) | 8 |
| SSM discretization range | $(10^{-3}, 10^{-1})$ |
| Latent variable | Multi-categorical |
| Categorical latent variable numer | 32 |
| Categorical classes number | 32 |
| Unimix probability | 0.01 |
| Learning rate | $10^{-4}$ |
| Reconstruction loss weight, $\beta_{\text{pred}}$ | 1 |
| Dynamics loss weight, $\beta_{\text{pred}}$ | 0.5 |
| Representation loss weight, $\beta_{\text{rep}}$ | 0.1 |
| World Model gradient clipping | 1000 |
| Adam epsilon | $10^{-8}$ |
| Actor Critic Hyperparameters | |
| Imagination horizon | 15 |
| Discount $\gamma$ | 0.997 |
| Return $\lambda$ | 0.95 |
| Entropy weight | $3 \cdot 10^{-4}$ |
| Critic EMA decay | 0.98 |
| Critic EMA regularizer | 1 |
| Return normalization scale | $\text{Per}(R, 95) - \text{Per}(R, 5)$ |
| Return normalization decay | 0.99 |
| Adam epsilon | $10^{-5}$ |
| Actor-Critic gradient clipping | 100 |

Table 3: Hyperparameter details for R2I. For an in-depth description, please refer to Section 3 and consult Hafner et al. (2023).

# F    BSUITE TRAINING CURVES

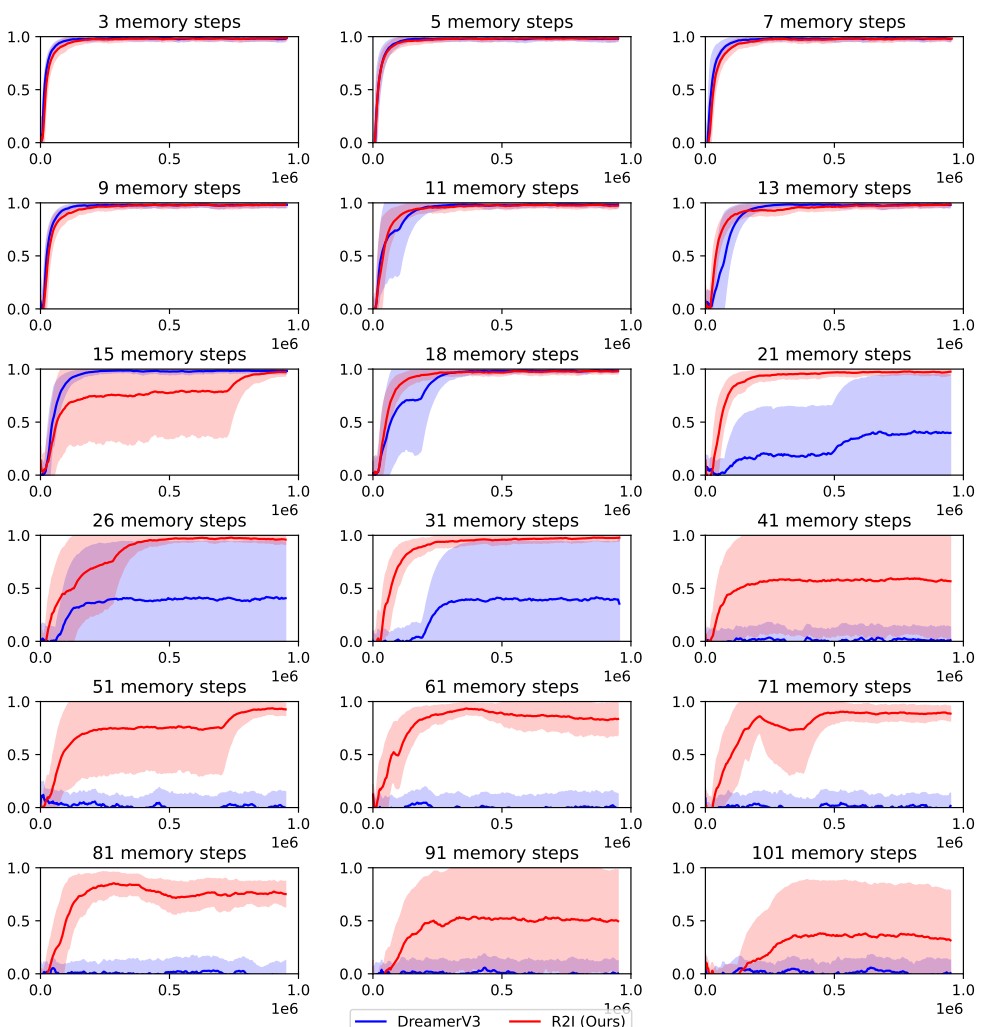

Figure 7: R2I and DreamerV3 training curves in BSuite `Memory Length` environment, averaged over 10 independent runs per environment (Median reward plotted).

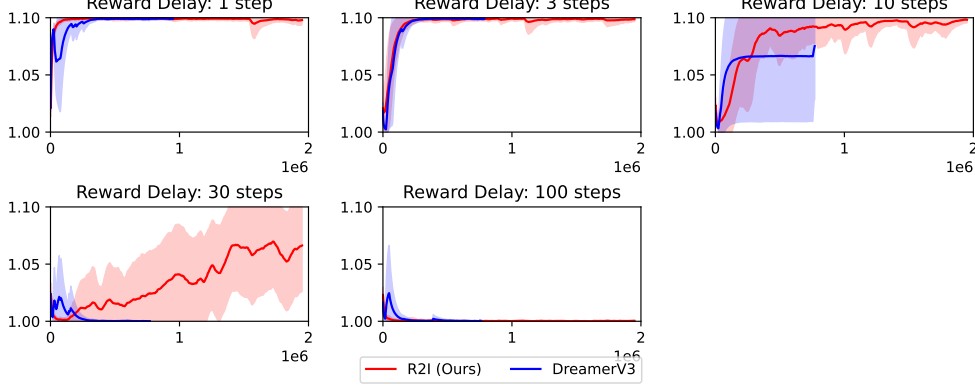

Figure 8: R2I and DreamerV3 training curves in BSuite `Discounting Chain` environment, averaged over 10 independent runs per environment (Median reward plotted).

# G POPGYM ENVIRONMENTS DETAILS

In this research, we examine three environments within the POPGym domain (Morad et al., 2023), each configured with `Easy`, `Medium`, and `Hard` levels of difficulty to systematically increase the memory demands for the agents for memory-intensive tasks. In those tasks, the fundamental challenge escalates with the difficulty level, necessitating the memorization and subsequent recall of a larger number of observations or actions for the more advanced tiers. Table 4 summarizes the different aspects of these tasks.

| | RepeatPrevious | | | Autoencode | | | Concentration | | |
|---|---|---|---|---|---|---|---|---|---|
| | -E | -M | -H | -E | -M | -H | -E | -M | -H |
| Parallel events to recall | 4 | 32 | 64 | 52 | 104 | 156 | 104 | 208 | 104 |
| The size of the observation space | | 4 | | | 4 | | 3 | 3 | 14 |
| Memory-less suboptimal policy | | ✗ | | | ✗ | | | ✓ | |

Table 4: Memory complexities of environments in POPGym. "Parallel events" refers to the maximum number of actions or observations that the agent needs to recall. The third row indicates whether the task can be partially solved without memory. `-E` is `-Easy`; `-M` is `-Medium`; `-H` is `-Hard`.

- `RepeatPrevious`. In this environment, both observation and action spaces are categorical with 4 discrete values. The task requires the agent to replicate an observation $a_{t+k} = o_t$ that occurred $k$ steps prior, at any given time step $t + k$. The agent at each time step needs to simultaneously remember $k$ categorical values; accordingly, the value of $k$ depends on the difficulty level, being 4 for `Easy`, 32 for `Medium`, and 64 for `Hard`, regardless of the fixed observation space size (which is 4).

- `Autoencode`. The environment operates in two distinct phases. In the "watch" phase, the environment generates a discrete observation from a set of 4 possible values at each time step; agent actions do not affect the observations during this phase. This phase spans the first half of the episode, lasting $T/2$ steps, where $T$ is the total length of the episode. The episode lengths are set to 104, 208, and 312 steps for the `Easy`, `Medium`, and `Hard` difficulty levels, respectively. In the subsequent phase, the agent must recall and reproduce the observations by outputting corresponding actions, which are drawn from the 4-value space, similar to the observation space. Essentially, the first step in the second phase should be equal to the first observation in the "watch" phase, the subsequent action in the second phase ought to be the same as the second observation from phase one, and so on. The agent receives both a phase indicator and the categorical ID to be repeated in its observation. Note that at each time step, the agent must remember $T/2$ categorical values; thus, successful completion of this task requires an explicit memory mechanism, and any policy performing better than random chance is utilizing some form of memory.

- `Concentration`. At each step, the agent receives an observation consisting of multiple categories, each with $N$ distinct values. The number of categories is 52 for `Easy` and `Hard`, and 104 for `Medium`. The values of $N$ are 3 for `Easy` and `Medium`, and 14 for `Hard`. The task simulates a card game with a deck of cards spread face down. The agent can flip two cards at each step, and if they match, they remain face up; otherwise, they are turned face down again. The agent's observation includes the state of the full deck, and the episode length corresponds to the minimal average number of steps required to solve the task optimally as determined by Morad et al. (2023). To outperform a random policy, the agent must remember card positions and values to find matches more efficiently. Even without memory, an agent can avoid flipping cards already turned face up, which conserves steps without yielding new information or reward, thereby outperforming random policies at a basic level. This is a marginal yet conspicuous improvement that shows a memory-less policy can gain over a random policy. Note that due to the episode length constraint, this task cannot be solved without memory, as proven by (Morad et al., 2023).

# H POPGYM TRAINING CURVES

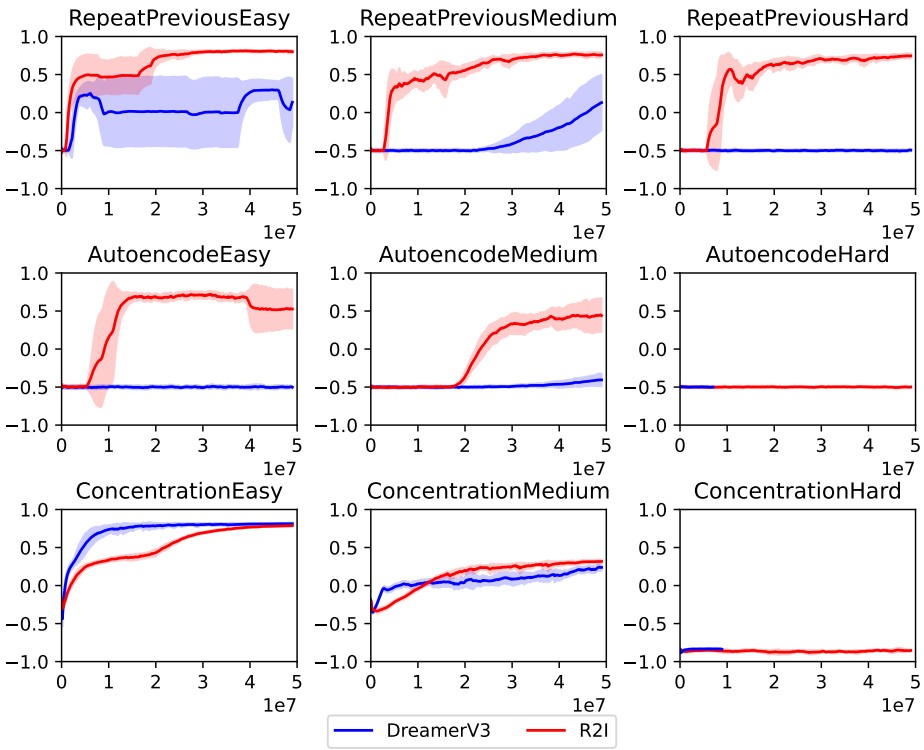

Figure 9: R2I and DreamerV3 training curves in the POPGym benchmark, averaged over 3 independent runs per environment.

# I    DMC-PROPRIO TRAINING CURVES

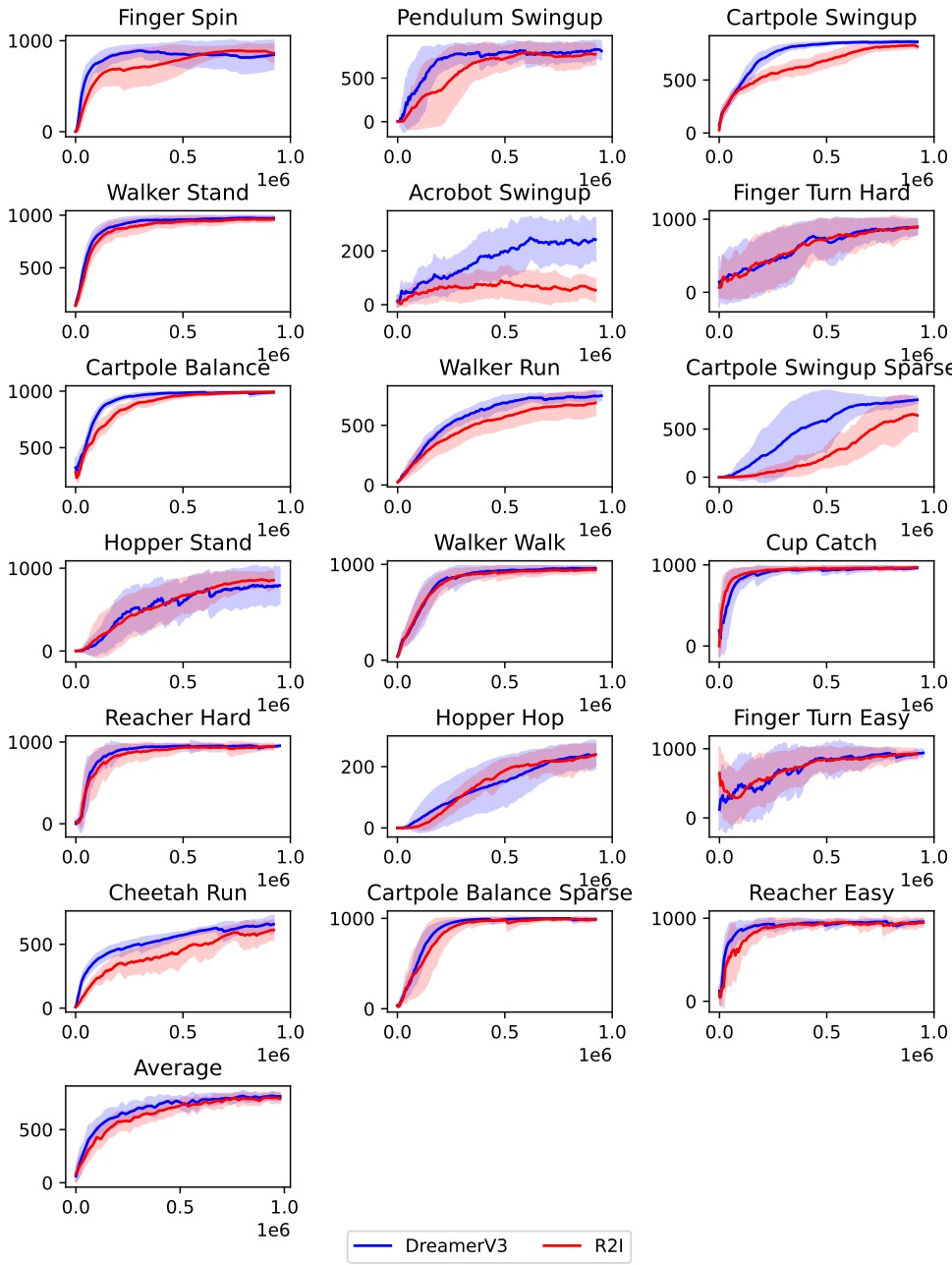

Figure 10: R2I and DreamerV3 training curves across proprioceptive environments in the DMC benchmark, averaged over 3 independent runs per environment.

## J  DMC-VISION TRAINING CURVES

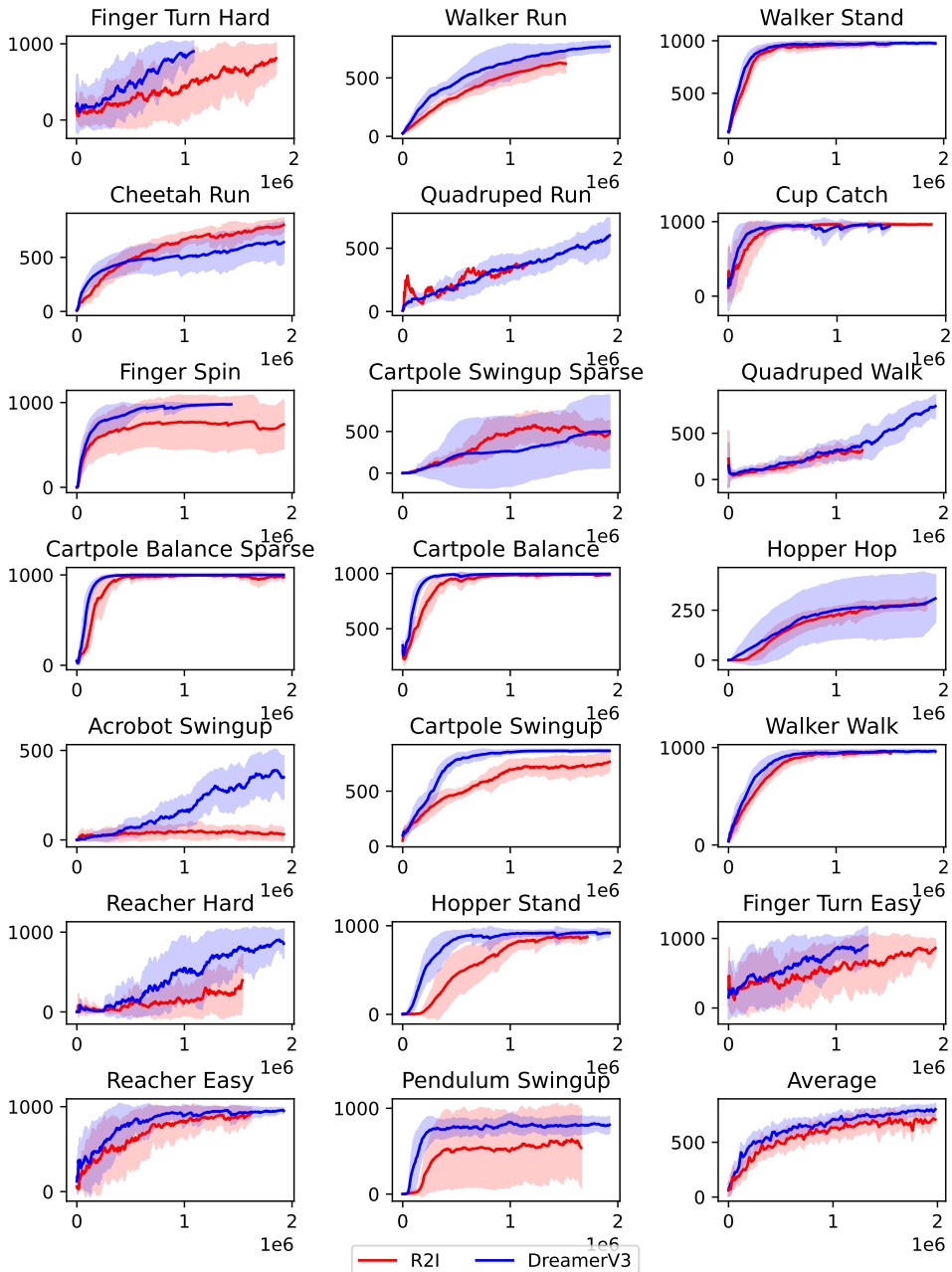

Figure 11: R2I and DreamerV3 training curves across visual environments in the DMC benchmark, averaged over 3 independent runs per environment.

# K    ATARI 100K TRAINING CURVES

Atari 100K benchmark (Łukasz Kaiser et al., 2020) is a standard RL benchmark comprising 26 Atari games featuring diverse gameplay mechanics. It is designed to assess a broad spectrum of agent skills, and agents are limited to executing 400 thousand discrete actions within each environment, which is approximately equivalent to 2 hours of human gameplay. To put this in perspective, when there are no constraints on sample efficiency, the typical practice is to train agents for 200M steps.

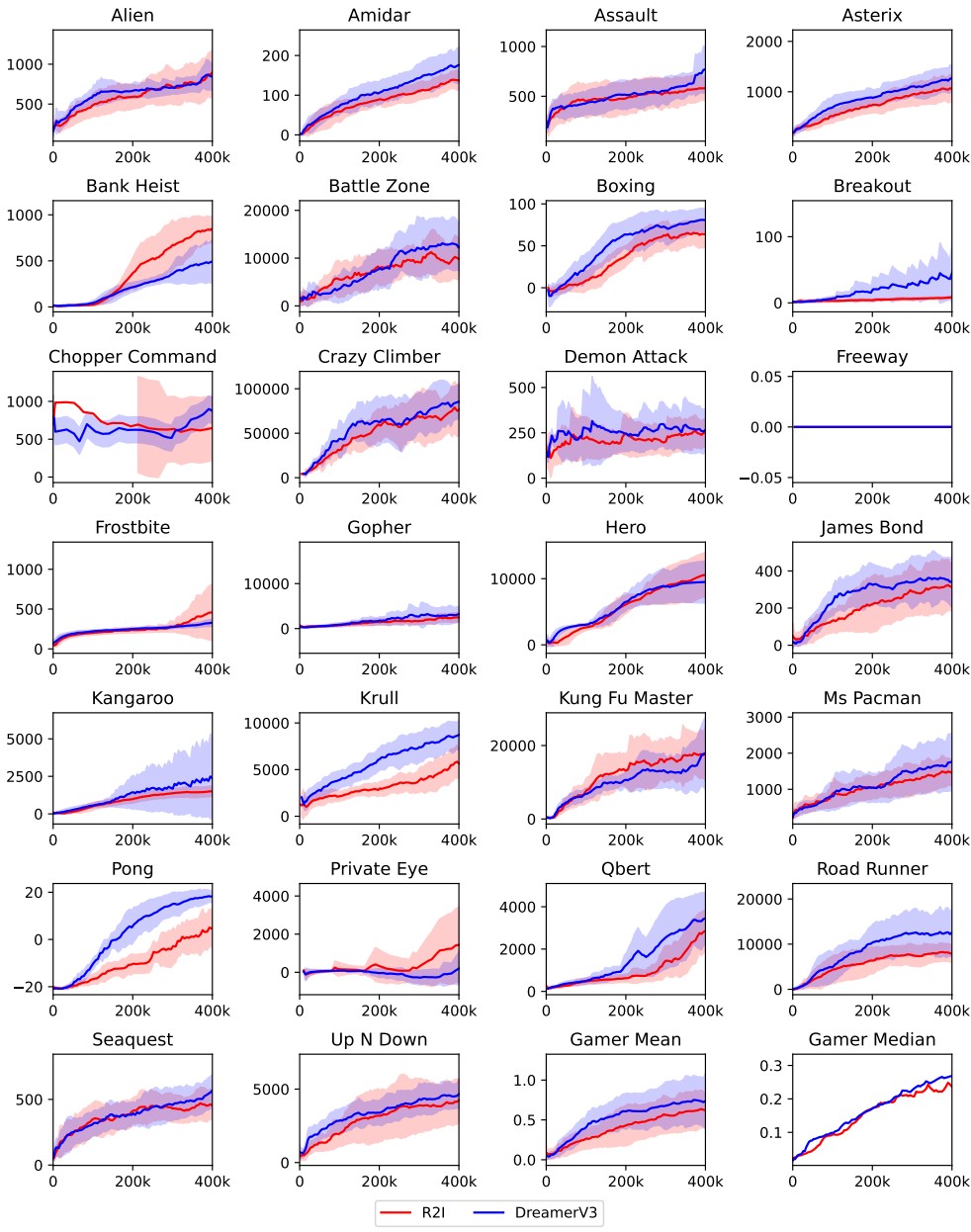

Figure 12: R2I and DreamerV3 training curves in the Atari after 400K environment steps, averaged over 3 independent runs per environment

## L    HYPERPARAMETER TUNING IN POPGYM

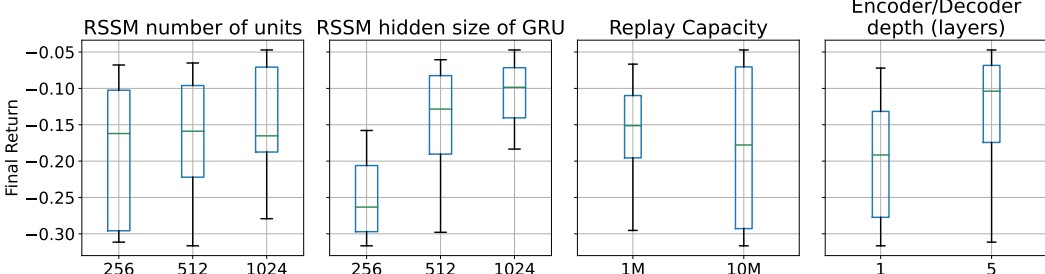

Figure 13: Results of hyperparameter tuning for DreamerV3 in POPGym memory environments.

To establish a robust baseline for comparison, we conducted an extensive hyperparameter search on the baseline DreamerV3 model. The results of this search are delineated in Figure 13. Each hyperparameter configuration of DreamerV3 was trained on 9 memory tasks from POPGym[1] considered in this study. To tune the model, we considered the following hyperparameters:

1. *Number of units in RSSM*, with the sizes of auxiliary layers ranging from $256, 512, 1024$;
2. *Hidden size of GRU in RSSM* with similar values;
3. *Replay buffer size* with two options, $10^6, 10^7$;
4. *Sizes of encoder and decoder.*

Each configuration used two random seeds over 10 million environment steps. The rationale underlying our choices is anchored in DreamerV3's origins as a general agent with all hyperparameters fixed except network sizes and training intensity, which control sample efficiency. To identify an optimal configuration, we tuned the model sizes around the configuration proposed for Behavior Suite (since POPGym and Behavior Suite share similar properties). We also checked an increased replay buffer size of 10M steps (in addition to the standard 1M of DreamerV3) since it improved the performance for R2I.

Figure 13 shows a box plot for each hyperparameter with the marginal performance of each hyperparameter (i.e., for hyperparameter **X**, we fix **X** but vary all other hyperparameters, forming an empirical distribution with median, max, min, etc.). We plot a box plot of the empirical distribution for each **X**. This visualization shows how much each hyperparameter can contribute to the overall performance (as it shows the median with all other hyperparameters varied but a chosen one fixed) and also what is the best hyperparameter value for the task (since the box plot shows the maximal value). Our results confirm the favorable scaling of DreamerV3 (Hafner et al., 2023) in this new environment — bigger networks learn bigger rewards. Therefore, we opt for $1024$ RSSM units, an RSSM hidden size of $1024$, and 5 layers of encoder/decoder. Unlike for R2I, for DreamerV3, a bigger replay performs less "stable" — since it has a bigger variation (e.g., increasing the network size might have an unexpected influence on the final agent performance). Yet, since with 10 million steps the model has the best score, we opt for this option for a fair comparison. The final DreamerV3 model reported in Figure 4 was trained on the best-found configuration.

---

[1]RepeatPrevious, Autoencode, Concentration for each difficulty: Easy, Medium, Hard

## M    IMPACT OF NON-RECURRENT REPRESENTATION MODEL

In this section, we study the ramifications of severing the link between the representation model and the sequence model. As previously discussed in Section 3, disrupting this connection facilitates the parallel computation of SSM sequence features (otherwise, if the representation model is fed by the sequence model's previous output, the whole computation must be sequential). Our investigation revolves around the impact of maintaining the recurrent pathway from the sequence model to the representation model.

First, we test how removing this recurrent connection affects the performance in the Memory Maze. We do so in two versions of this benchmark. The first one is the standard Memory Maze reported in this work. Subsequent to this, we explore an augmented variant of the benchmark that supplements the task with the coordinates of target objects, demanding the agent to process the visual input to reconstruct both the input image and the target positions (Pasukonis et al., 2022).

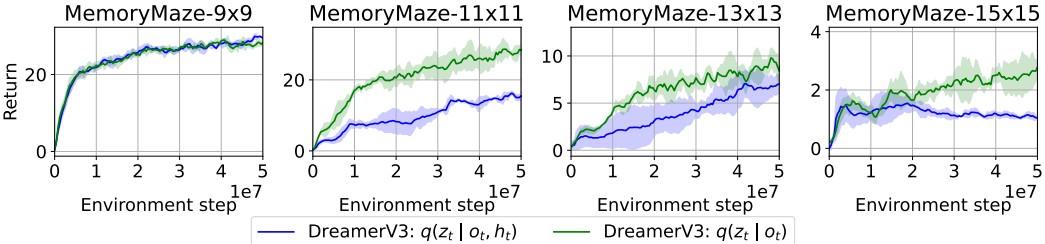

Figure 14: Representation model ablation in Memory Maze.

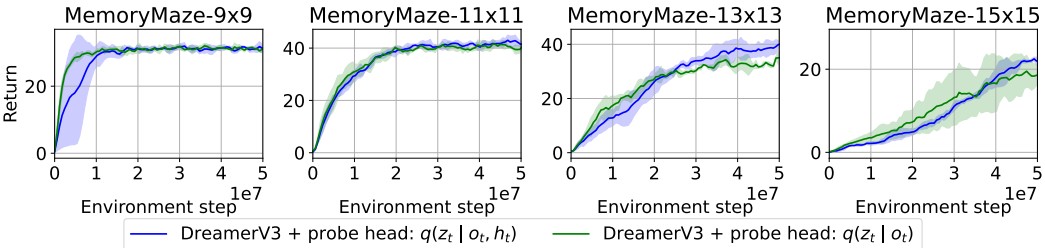

Figure 15: Representation model ablation in Memory Maze with probing head.

The empirical evidence, as depicted in Figures 14 and 15, suggests that the non-recurrent representation model retains or elevates performance levels. Note that the introduction of a probing network equips the agent with additional memory supervision, as it is tasked to predict target positions. Hence, in the absence of such memory supervision, the non-recurrent representation model demonstrates enhanced performance, while in its presence, it sustains performance without a loss.

In other words, the non-recurrent representation model is an inductive bias that favors memory. As shown in Figure 16, in standard RL benchmarks, this recurrent connection has zero influence on performance, confirming that this inductive bias favors memory without any negative side effects.

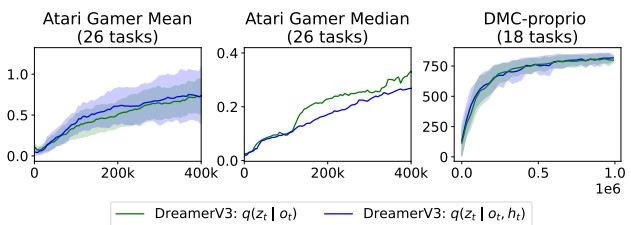

Figure 16: Representation models in standard RL environments.

## N    POLICY INPUT ABLATIONS

In this section, we aim to decipher which policy input yields superior performance. As a reminder, S3M comprises three main components: the deterministic output state $h_t$, leveraged by prediction heads to summarize information from previous steps; the stochastic state $z_t$, representing each single observation; and the hidden state $x_t$, which is passed between time steps. The policy variants are the output state policy $\pi(\hat{a}_t \mid z_t, h_t)$, the hidden state policy $\pi(\hat{a}_t \mid z_t, x_t)$, and the full state policy $\pi(\hat{a}_t \mid z_t, h_t, x_t)$.

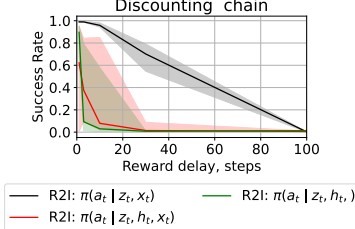

Figure 17: Actor-critic input ablations in `Discounting Chain`.

Our main insight is that in different environments $h_t$, $z_t$, and $x_t$ behave differently and they may represent different information. Additionally, the evolution of their distributions during world model training hurts the actor-critic training stability. Hence, we found each domain has its own working configuration. The only shared trait is that memory-intensive environments generally prefer policies that incorporate $x_t$, either through $\pi(\hat{a}_t \mid z_t, x_t)$ or $\pi(\hat{a}_t \mid z_t, h_t, x_t)$ – a finding corroborated by our analyses in Figures 17 and 18. In the specific context of the Memory Maze, the potency of input feature modification is smaller. As Figure 19 shows, the output state policy performs worse than the other two.

However, there exists an extended version of the Memory Maze benchmark. This task adds additional information which is the target objects' coordinates. Although the agent is still given only an image as an observation, it is now tasked to reconstruct both the input image and target positions (Pasukonis et al., 2022). As Figure 20 shares, illustrates, all three policy variants achieve comparable reward out-

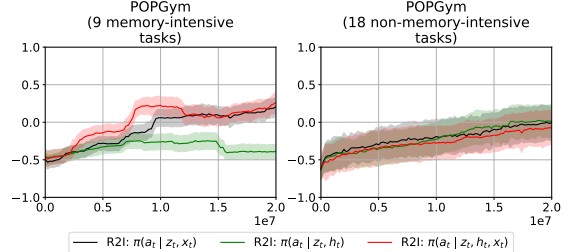

Figure 18: Actor-critic input ablations in POPGym.

comes, suggesting that the differentials between the input states—specifically ( $(z_t, x_t)$ and $(z_t, h_t)$) configurations—do not yield distinct advantages in this augmented task scenario.

Concluding our analysis, Figure 21 reveals a nuanced but noteworthy observation: the variations in policy inputs exert a marginal, albeit detectable, effect on the final performance. This implies a considerable overlap in the informational content encapsulated by $(z_t, h_t)$ and $(z_t, x_t)$.

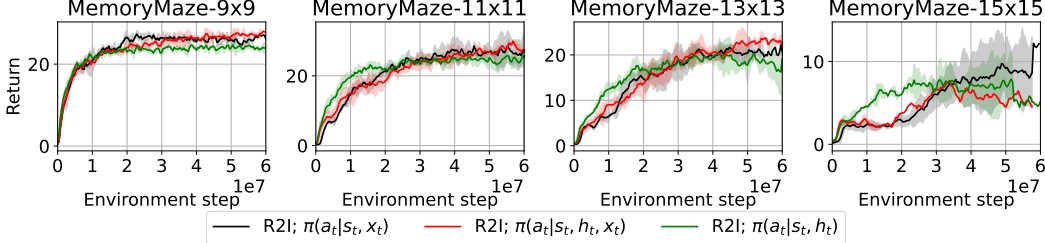

Figure 19: Memory Maze policy ablations without probing heads. The performances of all three variants are qualitatively different indicating that the information in the S3M output $h_t$ and S3M hidden $x_t$ is different. In addition, the output state policy $\pi(\hat{a}_t \mid z_t, h_t)$ exhibits a decline in performance as the training progresses.

---

[1]Non-memory-intensive environments include: `CountRecall`, `Battleship`, `MineSweeper`, `RepeatFirst`, `LabyrinthEscape`, `LabyrinthExplore`. All three difficulties (`Easy`, `Medium`, `Hard`) for each.

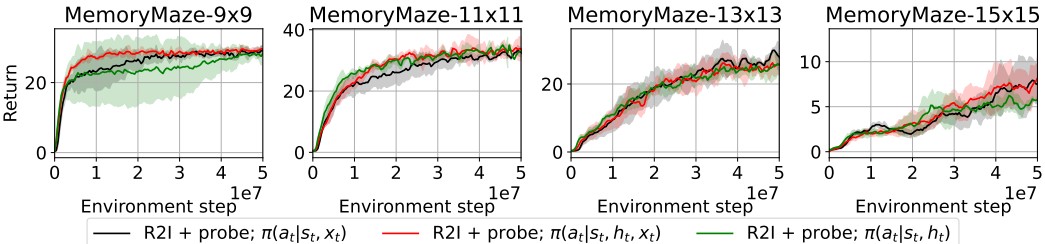

Figure 20: With probing head in Memory Maze, the agent is tasked to predict object locations from $h_t$ (S3M output), effectively making $h_t$ a Markovian state (i.e. equivalent to hidden state $x_t$). Thus, we can see the performance of all three policy variants is effectively the same.

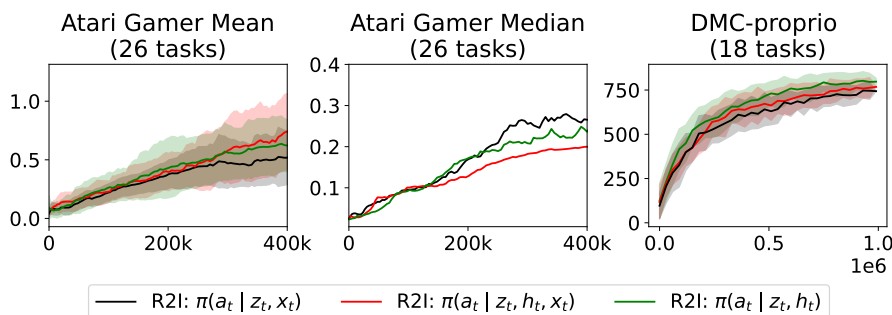

Figure 21: Actor-critic input ablations in Atari 100K and DMC domains.

## O  CRITICALITY OF FULL EPISODE INCLUSION IN TRAINING BATCH

In the domain of memory-intensive tasks, sequence length within training batches plays a pivotal role in the final performance. Our findings suggest that the success of our model is strongly contingent on accommodating the entire episode within a single sequence in the batch. This requirement ensures that the model can fully leverage temporal dependencies without the truncation of episodes. Thanks to parallel scan (Blelloch, 1990) (please refer to sections 2.1 and 3 for elaboration), the batch sequence length scales as efficiently as scaling the batch size. Therefore, we opt to train the model on complete episodes in the training batch across all experimental environments.

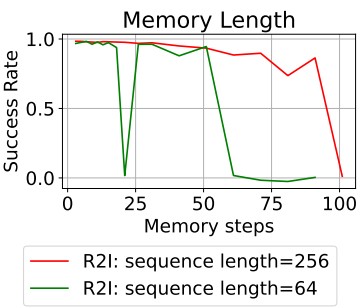

Figure 22: Ablation study of input sequence lengths for models on `Memory Length`.

In our experiments, depicted in Figure 22, we trained our model with two distinct sequence lengths in batch: 64 and 256 steps. As the episode length increases along the x-axis, the agent's ability to retain and utilize the initial observation to produce the correct action at the end is tested. The results indicate a clear performance dichotomy: when episodes surpass 64 steps, models trained with batch sequences of this length fail to perform (achieving zero performance), while models trained with 256-step sequences maintain robust performance, even as episode lengths extend. This suggests that encompassing the entire RL episode within the training batch is crucial for model efficacy.

As shown in Figure 23, the performance scales positively with the sequence length, reinforcing the finding above. Here, we test the R2I model across four sequence lengths in batch: 64, 128, 256, and 1024 steps, observing a positive correlation between sequence length and performance. Throughout this range, performance improvements are evident with each incremental increase in sequence length, underscoring the importance of longer sequences for successful memory task execution.

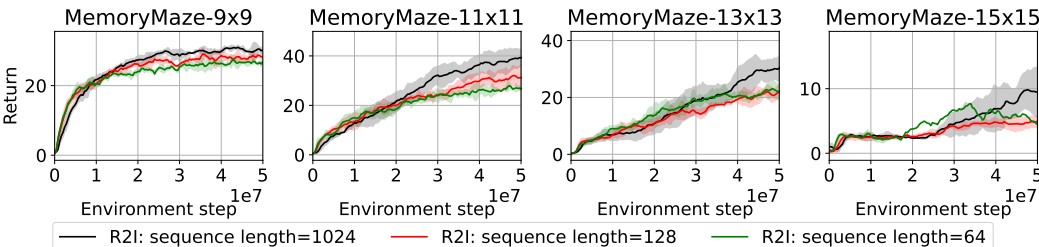

Figure 23: Ablation study of input sequence lengths for models on Memory Maze domain.

# P   MODEL LIKELIHOOD RELATION WITH AGENT PERFORMANCE

In MBRL, the agent typically goes through a process of learning a world model and employs it for planning or optimizing the policy. Therefore, one might naturally assume a higher likelihood of the world model leads to improved policy performance. Nevertheless, this assumption does not hold true in all cases. When the world model has constrained representational capacity or its class is misspecified, its higher likelihood may not necessarily translate into better agent performance (Joseph et al., 2013; Lambert et al., 2020; Nikishin et al., 2021). In other words, the inaccuracies in the world model will only minimally affect the policy if it happens to be perfect. Conversely, in the case of an imperfect model, these inaccuracies can lead to subtle yet significant impacts on the agent's overall performance (Abbad, 1991). For example, Nikishin et al. (2021) propose a method wherein the model likelihood is less than a random baseline while the agent achieves higher returns. This implies that the learned world model may not always have to lead to predictions that closely align with the actual states, which are essential for optimizing policies.

## P.1   COMPARING R2I WITH S4WM

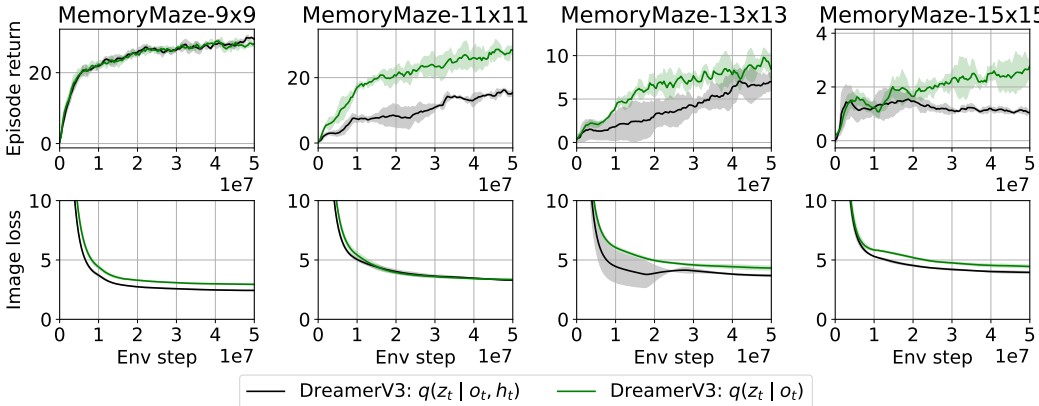

Figure 24: Online RL plots of the DreamerV3 model with recurrent and non-recurrent representation models (for details see Appendix M).

In parallel with our study, Deng et al. (2023) introduce a S4-based (Gu et al., 2021a) world model, called S4WM. The incorporation of S4 in the Dreamer world model has proven beneficial, as evidenced by the achievement of higher accuracy with lower Mean Squared Error (MSE). These results were obtained using offline datasets across various tabular environments and the Memory Maze (Pasukonis et al., 2022). However, their work primarily focuses on world modeling; for instance, there is no performance report in terms of obtained rewards. As previously discussed, a higher model likelihood does not necessarily guarantee better performance (i.e., higher rewards).

Figure 24 illustrates the results of DreamerV3 (which also serves as the predecessor to S4WM) with two distinct configurations for the world model on Memory Maze. One agent (depicted in green) incorporates a non-recurrent representation model, while the other (depicted in black) uses a recurrent one. Deng et al. (2023) utilize image loss as one of the key indicators for model likelihood. As shown, despite the agent with a non-recurrent representation model having a higher image loss, it manages to achieve higher rewards. This result is related to the experimental results of S4WM in the following manner. In S4WM, the authors conduct experiments with the datasets collected by scripted policies within environments like Memory Maze such as Four-rooms, Eight-rooms, and Ten-rooms. Also, they evaluate image generation metrics on a heldout dataset (compared to what the world model was trained on) to assess the in-context learning proficiency of the world model across unseen levels. Similarly, the outcomes depicted in Figure 24 provide online RL metrics within memory maze environments. It is noteworthy that here, the agent is trained with a replay buffer that is progressively updated. Therefore, successive batches may contain data from episodes within newly generated mazes that the model has not encountered before, reinforcing our assertion that these results are indeed comparable.

Table 5: Architectural comparison between R2I and S4WM

| Feature | R2I | S4WM |
|---|---|---|
| Computational modeling | Parallel scan | Conv mode |
| Layer normalization | Postnorm | Prenorm |
| SSM in posterior and prior | Shared | Not shared[3] |
| Post-SSM transformation | GLU transformation | Linear transformation |
| SSM dimensionality | MIMO | SISO |
| SSM parameterization | Diagonal | DPLR |
| SSM discretization | Bilinear | ? (Unspecified) |
| Handling resets technique | Reset to zero or learnable vectors | Not implemented |
| Exposure of hidden states | Provided by parallel scan | Not implemented |
| Policy training mode | DreamerV3's objective | None |
| World model objective | Same as DreamerV3 | Same as DreamerV3 but without free info in KL |

This highlights the idea that, in order to enhance long-term temporal reasoning in the context of MBRL and tackle memory-intensive tasks, simply maximizing world model likelihood is insufficient. That is why S4WM incorporates a completely different set of hyperparameters and design decisions, as detailed in Table 5. Here is a comprehensive explanation of each design decision that sets S4WM apart from R2I:

1. **Computational modeling.** R2I uses parallel scan (Blelloch, 1990; Smith et al., 2023) while S4WM uses global convolution (Gu et al., 2021a) for training. One difference between these two techniques is that parallel scan uses SSM to compute $u_{1:T}, x_0 \rightarrow y_{1:T}, x_{1:T}$ while convolution mode uses SSM to compute $u_{1:T}, x_0 \rightarrow y_{1:T}, x_T$. Yet, as discussed in Section 3.2 and empirically shown in Appendix N, providing the policy with the SSMs' hidden states $x_{1:T}$ is crucial for the agent's performance. This is not feasible with the convolution mode, as it does not yield these states.

2. **Layer normalization.** R2I applies post-normalization (i.e., after the SSM layers and right after the merge of the residual connections), whereas S4WM employs pre-normalization (i.e., before the SSM layer but subsequent to the residual branching). Pre-normalization tends to be more conducive for training models with significant depth, whereas post-normalization can offer a modest enhancement in generalization when a fewer number of layers are used (Wang et al., 2019). Given that our model must continuously generalize on data newly added to the replay buffer (as it performs online RL), post-normalization emerges as the more intuitive selection for Online RL. This choice is confirmed by our preliminary results.

3. **Shared SSM network.** R2I employs a shared SSM network for all prior prediction and posterior stochastic and deterministic models' inference, whereas S4WM investigates both shared and non-shared options, ultimately reporting the non-shared option as superior for image prediction. In this study, we have not explored the non-shared option; however, we anticipate that the non-shared version might prove less efficient for policy learning due to potential feature divergence between the posterior and prior SSM networks. This is because the policy is trained using features from the prior model but deployed in the environment with features from the posterior model.

4. **Post-SSM transformation.** R2I uses a GLU transformation (Dauphin et al., 2017), while S4WM uses a linear fully connected transformation. In our preliminary experiments, we explored both options and found that the GLU transformation yields better empirical results, particularly in terms of the expected return.

---

[1]Here we refer to the S4WM-FullPosterior model. Even though it is not the main model, the S4WM-FullPosterior one showed superior image prediction performance.

5. **SSM dimensionality.** R2I employs MIMO (Smith et al., 2023), while S4WM utilizes SISO (Gu et al., 2021a). Although the superiority of one over the other is not distinctly evident, MIMO provides more fine-grained control over the number of parameters in the SSM.

6. **SSM parameterization.** R2I adopts a diagonal parameterization (Gupta et al., 2022a) of the SSM, while S4WM employs a Diagonal-Plus-Low-Rank (DPLR) parameterization (Gu et al., 2021a). In our experiments, we found that both approaches perform comparably well in terms of agent performance. However, DPLR results in a significantly slower computation of imagination steps—approximately 2 to 3 times slower—therefore, we opted for the diagonal parameterization (See Appendix B.1.1).

7. **SSM discretization.** R2I utilizes bilinear discretization for its simplicity. S4WM does not specify the discretization method used; however, the standard choice for discretization within the S4 model typically relies on Woodbury's Identity (Gu et al., 2021a), which necessitates matrix inversion at every training step—a relatively costly operation. This is the reason we decided early on in this work not to proceed with S4.

8. **World model training objective.** R2I uses the same objective introduced in DreamerV3. In contrast, S4WM utilizes the ELBO objective in the formulation of DreamerV3, with the sole difference being that free information (Hafner et al., 2023) is not incorporated in the KL term. Notably, free information was introduced in DreamerV3 as a remedy to the policy overfitting (it is evidenced by the ablations in Hafner et al. (2023)).

9. **Episode reset handling technique.** R2I leverages the parallel scan operator introduced by Lu et al. (2024) and modifies it to allow a learnable or zero vector when an RL episode is reset, a technique that was introduced in DreamerV3. While S4WM is also based on the DreamerV3 framework, it overlooks this modification and lacks the mechanism to integrate episode boundaries into the training batch. This adaptation is critical under the non-stationarity of the policy being trained; it is primarily aimed at enhancing the efficiency of policy training; as the policy distribution changes, resulting in adjusted episode lengths, S4WM will be required to modify its sequence length accordingly—either increasing or decreasing it as necessary.

## Q  UNVEILING THE PERFORMANCE IN STANDARD RL BENCHMARKS

To facilitate a more comprehensive comparison between R2I and DreamerV3, we build performance profiles using the RLiable package (Agarwal et al., 2021). In particular, we use these profiles to illustrate the relationship between the distribution of final performances in R2I and DreamerV3, as depicted in Figure 25. In Atari 100K (Łukasz Kaiser et al., 2020), the performance profiles of DreamerV3 and R2I show remarkable similarity, indicating that their performance levels are almost identical. Looking at the proprioceptive and visual benchmarks in DMC (Tassa et al., 2018), we observe that R2I exhibits a drop in the fraction of runs with returns between 500 and 900. Outside of this interval, the difference between R2I and DreamerV3 diminishes significantly. Considering these observations, we conclude that R2I and DreamerV3 display comparable levels of performance across the majority of tested environments, aligning with our assertion that R2I does not compromise performance despite its enhanced memory capabilities. Note that this is a non-trivial property since these are significantly diverse environments, featuring continuous control (DMC), discrete control (Atari), stochastic dynamics (e.g., because of the sticky actions in Atari), deterministic dynamics (in DMC), sparse rewards (e.g., there are several environments in DMC built specifically with explicit reward sparsification), and dense rewards. Our findings confirm that the performance in the majority of these cases is preserved.

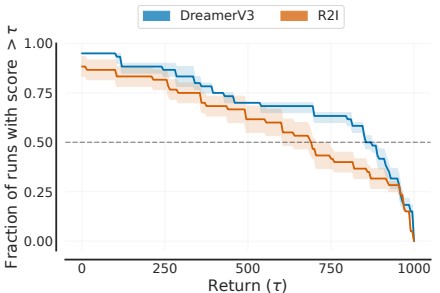

(a) Aggregated performance profiles for R2I and DreamerV3 across 20 visual environments in the DMC benchmark, averaged over 3 independent runs per environment.

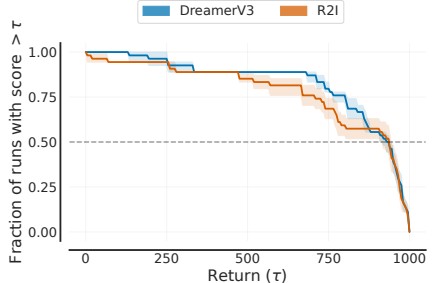

(b) Aggregated performance profiles for R2I and DreamerV3 across 18 proprioceptive environments in the DMC benchmark, averaged over 3 independent runs per environment.

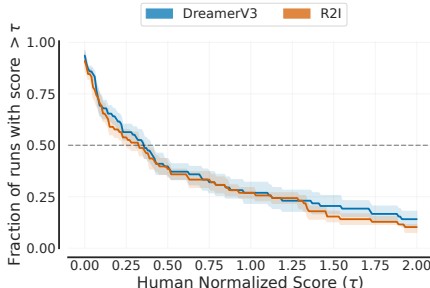

(c) Aggregated performance profiles for R2I and DreamerV3 across 26 environments in the Atari 100K benchmark, averaged over 3 independent runs per environment.

Figure 25: Comprehensive performance comparison profiles of DreamerV3 and R2I across three standard RL domains: Atari 100K, DMC-vision, and DMC-proprio. These visualizations highlight the comparative performance of the R2I and DreamerV3 across a diverse spectrum of tasks.

# R  ALGORITHM PSEUDOCODE

---

**Algorithm 1:** Recall to Imagine (R2I), full state policy training

---

Initialize empty FIFO Replay Buffer $\mathcal{D}$ ;
```
// P is set to the total env.  steps in batch to form at least
   one batch to start training
```
Prefill $\mathcal{D}$ with $P$ environment steps following the random policy;
**while** *not converged* **do**
    **for** *train step $c = 1..C$* **do**
```
        // World model training (parallel mode)
```
        Draw $n$ training trajectories $\{(a_t^j, o_t^j, r_t^j)\}_{t=k}^{k+L} = \tau_j \sim \mathcal{D}, \; j = \overline{1,n}$;
```
        // All variables are expected to include a batch dimension
        // which is omitted for simplicity
        // See Section 3.1
```
        Encode observations batch $z_{1:T} \sim q_\theta(z_{1:T} \mid o_{1:T})$;
        Compute S3M hidden states and outputs with parallel scan
        $h_{1:T}, x_{1:T} \leftarrow f_\theta((a_{1:T}, z_{1:T}), x_0)$;
```
        // Note that x_{1:T} include hidden states from all SSM layers
```
        Reconstruct rewards, observations, continue flags
        $\hat{r}_{1:T}, \hat{o}_{1:T}, \hat{c}_{1:T} \sim \prod_{t=1}^T p(o_t \mid z_t, h_t)p(r_t \mid z_t, h_t)p(c_t \mid z_t, h_t)$;
        Compute objective using Eq. (3), (4), (5), (6) and optimize WM parameters $\theta$;
```
        // Actor-critic training (recurrent mode)
        // See Section 2.2 and Appendix D for details
```
        Initialize imagination $x_{0|1:T}, h_{0|1:T}, z_{0|1:T} \leftarrow x_{1:T}, h_{1:T}, z_{1:T}$;
        **for** *imagination step $i = 0..H-1$* **do**
            $\hat{a}_{i|1:T} \sim \pi(\hat{a} \mid z_{i|1:T}, h_{i|1:T}, x_{i|1:T})$;
```
            // Note that this is a one-step inference of SSM in
            // recurrent mode
```
            $h_{i+1|1:T}, x_{i+1|1:T} \leftarrow f_\phi((\hat{a}_{i|1:T}, z_{i|1:T}), x_{i|1:T})$;
            $\hat{z}_{i+1|1:T} \sim p_\theta(\hat{z} \mid h_{i+1|1:T})$;
            $\hat{r}_{i+1|1:T} \sim p(r \mid \hat{z}_{i+1|1:T}, h_{i+1|1:T})$;
            $\hat{c}_{i+1|1:T} \sim p(c \mid \hat{z}_{i+1|1:T}, h_{i+1|1:T})$
        **end**
        Estimate returns $R_{1:H|1:T}^\lambda$ using Eq. 8 and $\hat{c}_{1:H|1:T}, \hat{r}_{1:H|1:T}$;
        Update actor-critic using estimated returns according to the rule from Hafner et al. (2023);
    **end**
    Collect $N$ steps in the environment using the World Model and the policy and store it to $\mathcal{D}$;
```
    // the ratio C/N is training intensity which should be
    // chosen according to the desired data-efficiency
```
**end**

---

Algorithm 1 presents the pseudo-code for training the R2I in an online RL fashion. Note that the imagination phase is initialized from every possible step in the training batch. For example, $\hat{h}_{1:H|1:T}$ is a tensor of shape `[batch_size, T, H, dim]`, which represents the imagination output after 1 to $H$ imagination steps starting from every possible time point (in total, there are $T$ of them). This concept was initially proposed by Dreamer (Hafner et al., 2019a) and subsequently adopted by DreamerV2 (Hafner et al., 2020) and DreamerV3 (Hafner et al., 2023). We employ the same idea since the hidden states of the SSM are outputted by the parallel scan algorithm, allowing the use of all hidden states to compute imagination steps.

## S  MODEL-FREE SSM IN POPGYM

Figure 4 presents the performance of both model-free and model-based approaches in the most memory-intensive environments in POPGym (Morad et al., 2023). In terms of model-based methodologies, Dreamer is included, while model-free baselines comprise Proximal Policy Optimization (PPO; Schulman et al. (2017)) executed with a variety of network architectures, such as MLP, GRU, LSTM, and S4D (Gu et al., 2022). S4D stands as an SSM approach characterized by a SISO framework (Gu et al., 2021a), a diagonal parameterization for its SSM, and an efficient convolution mode for parallelization. The primary distinction between S4 (Gu et al., 2021a) and S4D lies in their parameterization methods; S4D's diagonal approach, as opposed to S4's DPLR parameterization, which makes S4D faster and potentially more stable.

According to the findings reported in the POPGym paper (Morad et al., 2023), the `PPO+S4D` configuration emerged as the least performant among the thirteen model-free baselines. It frequently encountered numerical difficulties such as not-a-numbers (NaNs), which can be attributed to gradient explosions. Please note that due to S4D's reliance on convolution mode, it is incapable of managing multiple episodes within a sequence.

## T  DETAILED RESULTS IN MEMORY MAZE

The system achieves a total through-put of approximately 350 frames per second (FPS), leveraging two NVIDIA A100 GPUs with 40GB of memory, along with 40 environment workers. The training intensity is 51 replayed steps per 1 sampled step or 82 environment steps per one gradient update of the world model and actor-critic (since the batch size is 4096). To optimize utilization, training is dis-

| Env | Score | Human Normalized | Oracle Normalized | Steps | Wall time | A100 time |
|---|---|---|---|---|---|---|
| 9x9 | 33.55 | 127% | 96% | 17M | 0.6 | 1.2 |
| 11x11 | 51.96 | 117% | 89% | 66M | 2.2 | 4.4 |
| 13x13 | 58.14 | 104.7% | 78% | 206M | 6.9 | 13.8 |
| 15x15 | 40.86 | 60% | 46% | - | - | - |
| Avg | 46.13 | 102% | 77% | - | - | - |

Table 6: Algorithm statistics. Steps, Wall time, and A100 time are environment steps, wall time days, and A100 days until the superhuman performance.

tributed across the two A100 GPUs via batch-wise data parallelism. Additionally, we interleave training with environment sampling where the rollout worker lives on one of the two training GPUs.

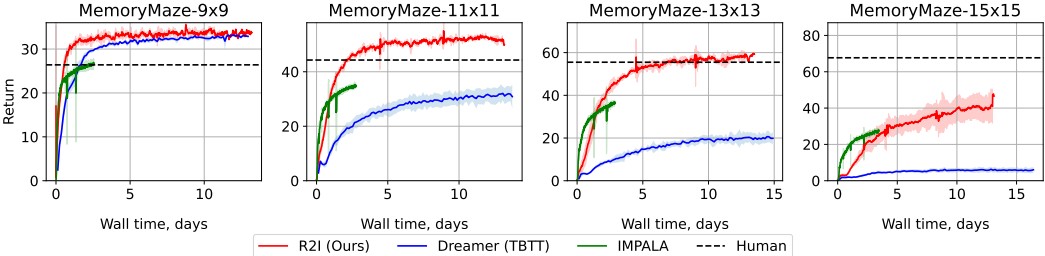

Figure 26: Wall time durations for training agents over several days in Memory Maze. Throughout the training period, Dreamer consistently requires equal or greater wall time compared to R2I, as its superior speed allows it to accumulate more environment steps within the same timeframe, as depicted in Fig 5.

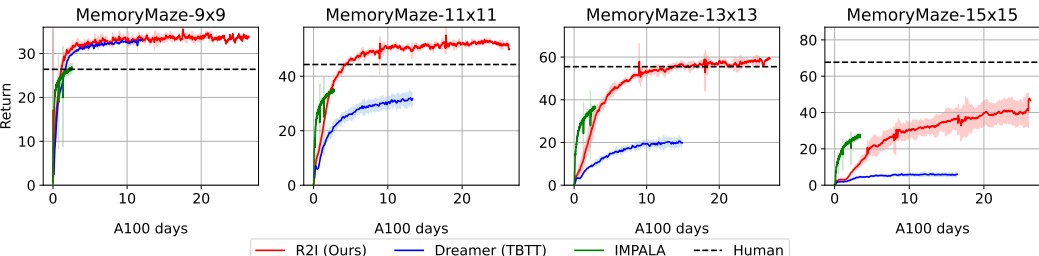

Figure 27: Training duration on A100 GPUs over several days.

