# OpenReview forum: "Mastering Memory Tasks with World Models"
_ICLR.cc/2024/Conference — ICLR 2024 oral_

### Official Review · Reviewer_fFhp · 2023-10-20

**Soundness:** 3 good
**Presentation:** 4 excellent
**Contribution:** 2 fair
**Rating:** 6
**Confidence:** 5

**Summary:**

This paper introduces state space models (SSM), in particular S4, into world models in the framework of model-based RL to improve its long-term memory and long-horizon credit assignment, as well as computational efficiency. Specifically, RSSM in Dreamer is replaced with SSM (S4), resulting in the proposed R2I agents. Design decisions to do so are carefully chosen, and empirical studies demonstrate improved performance in memory-demanding domains, including POPGym, bsuite, and Memory Maze.

**Strengths:**

1. Improved MBRL performance with S4-based world models has been validated in memory-demanding domains.
2. Careful designs of S4-based world models, including non-recurrent representation model and SSM computational modeling.
3. Extensive experiments in a number of domains.
4. Well written with sufficient experimental details.

**Weaknesses:**

1. Limited contribution. It is notable that there already exists a S4-based world model, namely S4WM [1]. Despite minor design choices, the major difference of this paper is that it conducts MBRL experiments while S4WM only conducts world model learning (e.g. imagination and reward prediction). However, in my humble opinion, it is not surprising that improvements in long-term memory can lead to improved MBRL performance in memory-demanding domains.
2. Three kinds of actor and critic inputs are introduced, namely, output state, hidden state and full state, which results in a critical design choice to be tuned for each domain. Although the authors provide some takeaways to select between them, it is not always true. For instance, output state policy is utilized in memory-demanding environments, Bsuite, while hidden state policy is used in non-memory environments, DMC.
3. The authors claim that R2I does not sacrifice generality for improved memory capabilities. However, there is a clear trend in Figure 6, that R2I performs worse than Dreamer in standard RL tasks.
4. Some inaccurate statements. For example, the authors say R2I's 'objective differs from ELBO in three ways', but to my knowledge, these three points are all borrowed from DreamerV3 but without explicitly being mentioned in the text.

[1] Deng et al. Facing off world model backbones: Rnns, transformers, and s4.

**Questions:**

The authors should properly resolve my concerns mentioned in the weakness part.

There are also some minor questions:

1. Dreamer is compared in Memory Maze tasks. Does this Dreamer baseline include the TBTT technique proposed by Pasukonis et al., which improves the memory of RSSM?
2. Why not include Transformer-based world models as baselines? Transformers are also widely believed to well model long-horizon dependencies.

---

> ### Author Response · Authors · 2023-11-16
> **Author Response**
>
> Thank you for your insightful feedback on our paper. We are encouraged by your recognition of the strengths in our work, including the improved performance of model-based RL with SSMs world models in memory-intensive settings, the careful design for non-recurrent representation and computational modeling, the breadth of our experimental validation across various domains, and the clarity and detail provided in our writing. We appreciate your comments and are eager to address your concerns and questions that you indicated.
>
> ---
>
> > Limited contribution. It is notable that there already exists a S4-based world model, namely S4WM.
>
> Indeed, another recent work that constructs an SSM-based world model, namely S4WM, appeared on Arxiv on **July 5th, 2023**, which is **2.5 months** before the ICLR paper deadline. However, we would like to highlight the fact that according to the ICLR public review guidelines, **all papers made available online within 4 months of the paper deadline are considered contemporaneous** (and authors are not required to compare with those papers). Despite this, we wish to provide an analytical comparison between R2I and S4WM, demonstrating that these are significantly different algorithms. We believe S4WM makes a valuable contribution to world modeling, which is a subtask of MBRL. **Improved world modeling quality is neither necessarily nor sufficiently linked to an improved expected reward in MBRL**. To support this claim, we have added a new section (please refer to the newly added **Section Q of the Appendix**) to the Appendix, with the following discussion: First, we present both theoretical and experimental evidence from existing literature that suggests the likelihood of the world model does not directly translate to reward [1, 2, 3], and sometimes a negative correlation is observed. Second, we conduct an experiment in Memory Maze environments, which is a task of primary interest in both S4WM and R2I, where we demonstrate cases of negative correlation between the return and image MSE (i.e., the expected return improves while image MSE worsens). We also interpret how this result relates to S4WM's experimental outcomes. Third, we provide a detailed table comparing all architectural choices made by R2I and S4WM, noting there are 9 non-trivial architectural differences, excluding network sizes and depths (although these also vary). For each difference, we interpret our understanding of its impact on policy learning. Please note that our comparison is with the S4WM version that was publicly available on Arxiv on July 5th, 2023, not the Arxiv version from November 9, 2023, and not the camera-ready version from NeurIPS 2023 (whose deadline was October 27, 2023)—one month after the ICLR paper deadline. Finally, S4WM reports metrics on offline RL datasets collected by agents assumed to be nearly optimal. However, in MBRL, the world model and policy commence training with data produced by a random agent. In summary, we believe R2I and S4WM are distinct models designed to solve different (but related) tasks, released concurrently online, with no evidence that S4WM will function out-of-the-box for reinforcement learning problems—on the contrary, there is evidence suggesting that significant modifications are necessary for S4WM to operate as an MBRL algorithm.
>
> ---
>
> > The authors claim that R2I does not sacrifice generality for improved memory capabilities. However, there is a clear trend in Figure 6, that R2I performs worse than Dreamer in standard RL tasks.
>
> We appreciate your attention to the details presented in the paper. We understand your concerns with the phrase “not sacrificing generality” and agree that our claim may benefit from moderation to more accurately reflect the experimental results. In the revision, we have adjusted our language to better communicate the nuanced performance outcomes (please refer to **Section 4.3**). Please also refer to the newly added **Appendix R** where we provide a more deep statistical analysis of the results on Standard RL tasks.
>
> While it's true that there are some tasks where DreamerV3 shows slightly better performance, our aggregated metrics suggest that the performance of DreamerV3 is preserved in the majority of environments. We employed the RLiable [4] package in the revision to show that the difference between them is negligible. It is worth noting that there are 64 such environments with diverse properties such as the type of control, reward sparsity, dynamics stochasticity, and more which is why preserving them all is extremely non-trivial.
>
> **References**
>
> [1] Joseph et al. Reinforcement learning with misspecified model classes. ICRA, 2013.
>
> [2] Lambert et al. Objective Mismatch in Model-based Reinforcement Learning. L4DC, 2020.
>
> [3] Nikishin et al. Control-oriented model-based reinforcement learning with implicit differentiation. AAAI, 2022.
>
> [4] Agarwal et al. Deep Reinforcement Learning at the Edge of the Statistical Precipice, NeurIPS 2021.

---

> ### Author Response · Authors · 2023-11-16
> **Author Response (Cont.)**
>
> > Three kinds of actor and critic inputs are introduced, namely, output state, hidden state and full state, which results in a critical design choice to be tuned for each domain. Although the authors provide some takeaways to select between them, it is not always true.
>
> Thank you for your feedback. We respectfully have a different view that we would like to clarify. To this end, we believe the policy input type should be chosen based on the type of partial observability exposed by the environment.
>
> As a reminder, the SSM network has a hidden vector \$x_t\$ (passing between steps of SSM) and output vector \$h_t\$ (passed to the observation and reward heads). For instance, consider the \$\texttt{Memory Length}\$ task in the BSuite. The agent observes the query vector in the first step and is then tasked to output an action corresponding to that query in the last step of the episode. The reward is only given in the last step of the episode and only in the case when the action aligns with the initial query. We therefore hypothesize that this provides an incentive for \$h_T\$ to represent everything necessary for the policy, which is why the output state policy works best there. In contrast, BSuite’s \$\texttt{Discounting Chain}\$ is a credit assignment problem where the rewards are delayed. Thus, we hypothesize that there is no such incentive for \$h_t\$ to represent everything needed by the policy, hence the hidden state policy works better.
>
> Lastly, despite DMC not being a memory task, the visual benchmark of DMC is a partially observable (PO) environment (since images do not showcase the velocity vectors). An empirical testimony for this fact (PO property of DMC-vision) can be found in the original DeepMind Control paper (see **Figure 6**). The same RL algorithm with an MLP or CNN backbone is applied to DMC environments with vector (full) and image (partial) observations. There exists a constant gap between these two variants, and the performance is lower for the image observation model. In the case of R2I, the best policy in DMC-proprio is the output state, since it is a fully observable environment and \$h_t\$ is tasked to predict this full observation. However, since DMC-vision is a PO domain, we found the hidden state policy to work better there.
>
> ---
> > The authors say R2I's 'objective differs from ELBO in three ways', but to my knowledge, these three points are all borrowed from DreamerV3 but without explicitly being mentioned in the text.
>
> We'd like to clarify that the integration of SSMs with DreamerV3's world model was explicitly stated at the beginning of **Section 3**. However, it is true that we did not explicitly mention that our objective is the same as in DreamerV3. Despite this, each time we mentioned any changes in the objective function in relation to ELBO, we properly cited DreamerV3 and other related works which initially proposed those changes. In order to avoid any future confusion, we have updated our paper to explicitly state that our objective is borrowed from the DreamerV3 algorithm.
>
> ---
> **Questions**
>
> ---
> > Dreamer is compared in Memory Maze tasks. Does this Dreamer baseline include the TBTT technique proposed by Pasukonis et al., which improves the memory of RSSM?
>
> Yes, to facilitate a more fair comparison, we included the TBTT version of the Dreamer in the initial submission. To avoid any future confusion, we have marked the Dreamer algorithm as **Dreamer (TBTT)**.
>
> ---
> > Why not include Transformer-based world models as baselines?
>
> Thank you for the suggestion to include Transformer-based world models as baselines. We did assess off-the-shelf Transformer models but found they performed poorly on desired tasks, without extensive hyperparameter tuning (which falls a bit outside the scope of this work). On another note, we also threw these models onto the evaluation for a speed test, which as expected, they were slower.
>
> ---
> **In conclusion, your constructive criticisms have been valuable in enhancing the quality and clarity of our research. We hope the revisions and clarifications provided have resolved your concerns. If your major questions and concerns have been addressed, we would appreciate it if you could support our work by increasing your score. If there are more questions/concerns, please let us know.**

---

> ### Comment · Reviewer_fFhp · 2023-11-17
>
> Many thanks for the detailed response!
>
> The authors have excellently addressed my concerns w.r.t. a concurrent work, S4WM.
>
> Despite that, I still think that introducing different kinds of actor and critic inputs, which need to be tuned or selected by your insights into specific environments, is a regretful deficiency.
>
> Moreover, a more deep statistical analysis actually supports the slightly inferior performance of R2I on standard RL benchmarks. Note that in the revision, the authors claim 'outside of this interval (500-900), the difference between R2I and DreamerV3 diminishes significantly' and conclude that 'R2I and DreamerV3 display comparable levels of performance
> across the majority of tested environments.' In contrast, there is a clear gap (\~10%) in the interval of 0~500, on the DMC-vision domain (Fig. 25a).
>
> Overall, I think this paper is worth accepting, and I decide to change my score to 6. I highly encourage the authors to consider my latest feedback and refine their claims further.

---

> ### Author Response · Authors · 2023-11-21
> **Thank you!**
>
> We sincerely appreciate increasing the score for our paper and your recognition of our efforts in addressing the concerns.
>
> We understand your point about the performance gap between R2I and DreamerV3. We have taken your remarks into serious consideration and would like to clarify the performance narrative based on comprehensive evaluations across various environments. Specifically, DreamerV3 and R2I are on par in 41 environments **(1)**. Our definition of "on par" is that the median (for a fixed game) performance of one method is within the confidence interval of the other algorithm, and **vice versa**. At the same time, there are 15 environments where R2I and DreamerV3 are insignificantly different **(2)**. Our definition of "insignificantly different" is that only one algorithm's median lies within the confidence interval of the other, while the other's median performance may lie outside of the first one algorithm - upper or lower. This is so because there are multiple different cases for the relation of scores between R2I and DreamerV3. Finally, if the median performance of R2I lies outside of DreamerV3's confidence, and DreamerV3's median performance lies outside of R2I's confidence, we say that they are significantly different. For R2I and DreamerV3, there are only 8 such environments **(3)**, and in one of them (Bank Heist of Atari 100K), R2I is significantly better than DreamerV3.
>
> In summary, there are 41 + 15 + 1 = 57 environments where R2I is not significantly worse than DreamerV3, and in the majority of these environments (41 out of 57), they perform on par. Out of all environments, the remaining 7 are the ones where DreamerV3 is significantly better than R2I, which is only approximately **10%** of all standard RL tasks considered (7/64).
>
> In light of these observations, we believe the generality claim in the paper is still correct. We propose to add the following to the text: “While R2I outperforms DreamerV3 in certain environments and vice versa, the overall performance analysis suggests that they offer similar behavior across a wide array of standard RL benchmarks, differing in only a small subset of these.”
>
> Would this articulation align with your insights, or would you advise further refinement? We appreciate any further guidance you may offer to help us align our revisions with your suggestions.
>
> ---
> **P.S.** We believe the issue of convincing the community that a certain method is not different from some other method comes from the lack of statistical methodology for RL. There exists only one notable work [1] which mainly focuses on determining if the methods are different but not if the methods perform comparably (these two questions assume different null hypotheses, and statistically answering one does not statistically answer the other). We are aware that certain statistical tests exist that answer questions similar to ours; however, we believe that a certain adaptation of these methods should be done to apply them to report results in RL.
>
>
> **(1)** Environments are DMC-proprio: Finger Spin, Pendulum Swingup, Walker Stand, Finger Turn Hard, Walker Run, Cartpole Balance, Cup Catch, Walker Walk, Hopper Stand, Reacher Hard, Hopper Hop, Finger Turn Easy, Cheetah Run, Cartpole Balance; DMC-vision: Walker Stand, Quadruped Run, Cup Catch, Cartpole Swingup Sparse, Quadruped Walk, Cartpole Balance Sparse, Cartpole Balance, Hopper Hop, Walker Walk, Finger Turn Easy, Reacher Easy; Atari100k: Alien, Asterix, Battle Zone, Boxing, Crazy Climber, Demon Attack, Freeway, Gopher, Hero, James Bond, Kung Fu Master, Ms Pacman, Qbert, Seaquest, UpNDown.
>
> **(2)** Environments are DMC-proprio: Cartpole Swingup, Cartpole Swingup Sparse; DMC-vision: Finger Turn Hard, Cheetah Run, Finger Spin, Finger Turn Easy, Pendulum Swingup; Atari100k: Amidar, Assault, Breakout, Chopper Command, Frostbite, Kangaroo, Private Eye, Road Runner.
>
> **(3)** Environments are DMC-proprio: Acrobot Swingup; DMC-vision: Walker Run, Acrobot Swingup, Cartpole Swingup, Reacher Hard;  Atari100k: Bank Heist, Krull, Pong.
>
>
> ---
> **Reference**
>
> [1] Agarwal et al. Deep Reinforcement Learning at the Edge of the Statistical Precipice, NeurIPS 2021.

---

### Official Review · Reviewer_pPqM · 2023-10-23

**Soundness:** 3 good
**Presentation:** 3 good
**Contribution:** 3 good
**Rating:** 8
**Confidence:** 4

**Summary:**

This work proposes Recall to Imagine (R2I), a Model-Based Reinforcement Learning (MBRL) agent that integrates the Dreamer framework with State Space Models (SSMs), in order to alleviate the well-known challenges of long-term dependencies regarding memory and credit assignment. This integration works by replacing the GRU-based representation model with the SSM, which enables parallel predictions and improves the capabilities for capturing long-term dependencies. The paper presents a rich empirical analysis in a variety of memory environments, achieving state-of-the-art results in them while incurring a small drop in performance in the standard benchmarks. Furthermore, the paper provides an extensive ablation analysis of diverse design choices and hyperparameters in this integration.

**Strengths:**

- The paper addresses the problem of long-term dependencies for World Models, which is an issue for RNNs and Transformers on handling sequences in representational models. Therefore, it is very relevant to the community.

    - Furthermore, employing SSMs to replace the aforementioned backbones for learning temporal dependencies is sound and well-motivated.

- The proposed architecture establishes new state-of-the-art performance for several tasks in the considered environments (BSuite, POPGym, and Memory Maze), with a noticeable improvement in computational efficiency, as shown in Figure 2.

- The work brings extensive and insightful ablation studies in many design decisions in the R2I architecture. These are presented as a systematic evaluation in Appendices M to P and helps understanding how the proposed method works.

**Weaknesses:**

- Despite the conducted ablation in Appendix O, the question on why the different input variations work differently across environments still remains open. And the raised hypothesis on “feature instability” sounds vague and not properly backed up by a solid argument. It would be great to provide a better understanding of this challenge to give more clarity on how the method works, but I understand this is a difficult open problem that demands a careful investigation.

- The work adopts SSMs to address the challenge of handling long-term dependencies in RL (memory and credit assignment). Nevertheless, it does not motivate the employment of Dreamer (or, more generally, Model-Based RL). I think it is important to describe and motivate why Dreamer was used instead of Model-Free RL, as it is not clear why MBRL would be better than MFRL for these memory tasks (unless there is another motivation besides asymptotic performance, such as sample efficiency).

    - In the same line, Figure 4 brings some “memory-augmented” Model-Free baselines, but it lacks “PPO + SSMs”. This baseline would definitely clarify my concern. If there is no constraint in the sample budget, it is possible (perhaps expected) that the MFRL agent would perform better.

- In Section 4.3, the claim of “not sacrificing generality” is questionable. There is a small drop in performance. For instance, in Appendix J (DMC-proprio), DreamerV3 is (at least) slightly better in 6 tasks.  In Appendix K (DMC-Vision), 8 environments. In Appendix L (Atari), 10 environments. I suggest rephrasing the claim to account what is observed in the Appendices.

- The work from Deng et.al [1], proposing S4WM, looks very similar to the proposed one. Indeed, both works propose replacing the RSSM with SSMs with the same motivation: improving memory capabilities. The work on S4WM was publicly released approximately 2.5 months before this submission, which can be seen as concurrent work. Nevertheless, I believe the work is almost overlooked by the proposed paper, which has a small citation in Section 5. Given the similarity, it would be crucial to provide a more detailed comparison contrasting both works, in terms of methodology and evaluation, perhaps in the Introduction or in a separate Appendix.

**Minor Concerns**

- In Appendix H, task Autoencode: the episode length for the Hard task is 156. Is that right? I believe it is supposed to be 256.

**References**

[1] Deng et. al. Facing off World Model Backbones: RNNs, Transformers, and S4. NeurIPS, 2023.

**Summary of the Review**

The work brings an effective improvement on World Models due to a well-motivated employment of State Space Models. This validates this architecture and extends its effectiveness in dealing with sequences in RL. The work does a great job in empirical analysis, anticipating many questions and answering them with extensive experiments and ablations. On the other hand, I believe the paper could be improved if the aforementioned concerns were addressed. Nevertheless, these concerns are not critical enough to prevent acceptance.

**Questions:**

- In Appendix G (BSuite environment), is there any hypothesis on why sometimes harder environments (longer memory steps) present better performance than easier ones? For instance, R2I’s performance on 31 memory steps is better than 15 memory steps. Similarly, the performance in 81 memory steps seems better (or more stable) than 41 memory steps.










===================== **POST-REBUTTAL** ==================================

Dear authors,

Thanks for putting so much effort on addressing my concerns. I believe they led to substantial improvements in an already good work, so I am raising my scores towards acceptance.

Specifically:

- I appreciate the efforts on formulating hypotheses on why the different input variations work differently across environments. I believe the raised hypotheses are valid and potential venues for future work. As I mentioned before, this seems to be a difficult problem that requires careful investigation. It would be interesting to bring this rebuttal discussion into the paper, as I believe this could also be an important question for other readers. This is also valid to the point related to Appendix G.

- I also would like to thank you for adding the new PPO + SSMs baseline. Turns out to be very different from what I expected, and given the discussion on Appendix T, I think it should require careful tuning to work with PPO (which is, indeed, a very sensible algorithm). But this is an argument in favor of the proposed method and perhaps another open question to be addressed (i.e., how to make SSM to work with PPO)

- Thanks for addressing the tone of the claim in Section 4.3 and providing more evidence regarding it. I agree that given the diversity of environments, it is hard to ensure that all of them will attain the same performance, and the new wording also better reflects the presented results.

- Lastly, the Appendix Q is great. I think this was one common concern from many reviewers and it was well addressed with many details. For sure the strongest reason to increase the scores.

---

> ### Author Response · Authors · 2023-11-16
> **Author Response**
>
> Thank you for the constructive feedback on our submission. We appreciate your recognition of our approach to handling long-term dependencies in world models using SSMs and are pleased to note the acknowledgment of our extensive ablation studies. The new state-of-the-art performance and improved computational efficiency are key contributions of our work, which we are glad to have been highlighted. We believe these establish the significance and practical value of our research.
>
> We have carefully considered the concerns raised and would like to address them as follows:
>
> ---
> > Despite the conducted ablation in Appendix O, the question on why the different input variations work differently across environments still remains open.
>
> We acknowledge the reviewer's point on different policy inputs. To address this, we believe the policy input type should be chosen based on the type of partial observability exposed by the environment.
>
> As a reminder, the SSM network has a hidden vector \$x_t\$ (passing between steps of SSM) and output vector \$h_t\$ (passed to the observation and reward heads). For instance, consider the \$\texttt{Memory Length}\$ task in the BSuite. The agent observes the query vector in the first step and is then tasked to output an action corresponding to that query in the last step of the episode. The reward is only given in the last step of the episode and only in the case when the action aligns with the initial query. We therefore hypothesize that this provides an incentive for \$h_T\$ to represent everything necessary for the policy, which is why the output state policy works best there. In contrast, BSuite’s \$\texttt{Discounting Chain}\$ is a credit assignment problem where the rewards are delayed. Thus, we hypothesize that there is no such incentive for \$h_t\$ to represent everything needed by the policy, hence the hidden state policy works better.
>
> ---
> > It does not motivate the employment of Dreamer (or, more generally, Model-Based RL).
>
> > In the same line, Figure 4 brings some “memory-augmented” Model-Free baselines, but it lacks “PPO + SSMs”. This baseline would definitely clarify my concern.
>
> Thank you for pointing out the need for a more explicit rationale regarding our choice of MBRL and Dreamer. In response to your request, we wish to explain our motivation for concentrating on MBRL over MFRL. Our focus on MBRL stems from its potential in sample efficiency, reusability, transferability, and safer planning, along with the advantage of more controllable aspects due to its explicit supervised learning component (i.e., world modeling).
>
> In line with your feedback, we have integrated the “\$\texttt{PPO + SSMs}\$” baseline into our evaluations in POPGym (please refer to **Section 4.1** and **Figure 4**). All empirical results from our experiments on POPGym and Memory Maze speak to the aforementioned advantages of MBRL, suggesting the superiority of MBRL in these domains. Within the realm of MBRL methodologies, we decided to build upon the SOTA DreamerV3 framework, which has proven capacities for generalization, sample efficiency, and scalability.
>
> We acknowledge the need for an explicit comparison and motivation and have accordingly addressed this in the revised version of our paper in **Appendix B**, highlighting the rationale for choosing MBRL.
>
> ---
> > In Section 4.3, the claim of “not sacrificing generality” is questionable. There is a small drop in performance.
>
> We appreciate your attention to the details presented in the appendix. We understand your concerns with the phrase “not sacrificing generality” and agree that our claim may benefit from moderation to more accurately reflect the experimental results. In the revision, we have adjusted our language to better communicate the nuanced performance outcomes (please refer to **Section 4.3**). Please also refer to the newly added Appendix **section R** where we provide a more deep statistical analysis of the results on Standard RL tasks.
>
> While it's true that there are some tasks where DreamerV3 shows slightly better performance, our aggregated metrics suggest that the performance of DreamerV3 is preserved in the majority of environments. We employed the RLiable [1] package in the revision to show that the difference between them is negligible. It is worth noting that there are 64 such environments with diverse properties such as the type of control, reward sparsity, dynamics stochasticity, and more which is why preserving them all is extremely non-trivial.
>
> ---
> > In Appendix H, task Autoencode: the episode length for the Hard task is 156. Is that right? I believe it is supposed to be 256.
>
> Thank you for your careful reading of Appendix H. We have double-checked our experimental settings and can verify that the episode length specified for Autoencode-Hard is correct at 156. This is because the lengths in the Autoencode environments are multiple of 52 (the number of cards in a standard deck).

---

> ### Author Response · Authors · 2023-11-16
> **Author Response (Cont.)**
>
> > Given the similarity, it would be crucial to provide a more detailed comparison contrasting both works, in terms of methodology and evaluation, perhaps in the Introduction or in a separate Appendix.
>
> Thank you for suggesting a thorough comparison between R2I and S4WM. We understand the importance of distinguishing our contribution within the field, especially when similarities are present. To address this concern, we wish to provide an analytical comparison between R2I and S4WM, demonstrating that these are significantly different algorithms. We believe S4WM makes a valuable contribution to world modeling, which is a subtask of MBRL. **Improved world modeling quality is neither necessarily nor sufficiently linked to an improved expected reward in MBRL**. To support this claim, we have added a new section (please refer to the newly added **Section Q of the Appendix**) to the Appendix, with the following discussion: First, we present both theoretical and experimental evidence from existing literature that suggests the likelihood of the world model does not directly translate to reward [2, 3, 4], and sometimes a negative correlation is observed. Second, we conduct an experiment in Memory Maze environments, which is a task of primary interest in both S4WM and R2I, where we demonstrate cases of negative correlation between the return and image MSE (i.e., the expected return improves while image MSE worsens). We also interpret how this result relates to S4WM's experimental outcomes. Third, we provide a detailed table comparing all architectural choices made by R2I and S4WM, noting there are 9 non-trivial architectural differences, excluding network sizes and depths (although these also vary). For each difference, we interpret our understanding of its impact on policy learning. Please note that our comparison is with the S4WM version that was publicly available on Arxiv on July 5th, 2023, not the Arxiv version from November 9, 2023, and not the camera-ready version from NeurIPS 2023 (whose deadline was October 27, 2023)—one month after the ICLR paper deadline. Finally, S4WM reports metrics on offline RL datasets collected by agents assumed to be nearly optimal. However, in MBRL, the world model and policy commence training with data produced by a random agent. In summary, we believe R2I and S4WM are distinct models designed to solve different (but related) tasks, released concurrently online, with no evidence that S4WM will function out-of-the-box for reinforcement learning problems—on the contrary, there is evidence suggesting that significant modifications are necessary for S4WM to operate as an MBRL algorithm.
>
> ---
> **Questions**
>
> ---
> > In Appendix G (BSuite environment), is there any hypothesis on why sometimes harder environments (longer memory steps) present better performance than easier ones?
>
> While our paper did not explicitly hypothesize about the reasons behind this particular phenomenon, we can suggest a plausible explanation. **The observed behavior could be linked to the performance patterns of the DreamerV3 algorithm**; notably, at episode lengths 21, 26, and 31, DreamerV3 encounters similar challenges—this is evidenced by the variance arising from a subset of runs failing to converge to a substantial reward. Consequently, the distribution of final success rates for DreamerV3 exhibits bimodality in these instances, with certain final success rates achieving 1, while others remain at 0. R2I inherits this property of DreamerV3 but, due to its extended memory capabilities—which result in a higher number of successful cases—experiences greater fluctuations in the median success rate. This is the reason we opted for 10 seeds per episode length.
>
> To potentially resolve the issues, we can increase the number of seeds (which currently stands at 10). Stabilizing can be a good potential future direction to explore in R2I and Dreamer. Note that such instability has not been observed in any other environment tested in this study, with either DreamerV3 or R2I.
>
> ---
> **To wrap up, we appreciate your insightful observations and have diligently applied them to better our work. We hope that the revisions and elaborations made meet the high standards expected by the conference. If we have adequately responded to your key concerns and questions, we kindly ask for your consideration in enhancing the score you've allotted to our submission. However, if you still have any additional queries or concerns, we invite you to share them with us.**
>
> ---
> **References**
>
> [1] Agarwal et al. Deep Reinforcement Learning at the Edge of the Statistical Precipice, NeurIPS 2021.
>
> [2] Joseph et al. Reinforcement learning with misspecified model classes. ICRA, 2013.
>
> [3] Lambert et al. Objective Mismatch in Model-based Reinforcement Learning. L4DC, 2020.
>
> [4] Nikishin et al. Control-oriented model-based reinforcement learning with implicit differentiation. AAAI, 2022.

---

> > ### Comment · Reviewer_pPqM · 2023-11-22
> > **Thanks for the efforts during rebuttal**
> >
> > Dear authors,
> >
> > Thank you for the efforts during the rebuttal. I believe the rebuttal discussion and updates in the work addressed my concerns, and I am raising my score. Please check the updated review.

---

> > > ### Author Response · Authors · 2023-11-22
> > > **Author Response**
> > >
> > > Thank you for appreciating our responses and the updates made to the paper. Additionally, we express gratitude for the increased score. We will surely incorporate the feedback from you and other reviewers into the next revision of the paper (i.e., the camera-ready version, in the case of paper acceptance). In particular, we will add recommendations for practitioners wishing to work with our model (or similar models) regarding the choice of the policy input representation type and highlight the need for careful investigation of that in future work. Also, we will incorporate our intuition about the roots of the observed behavior of R2I and DreamerV3 (in bsuite), highlighting the direction for future work with both models. We additionally thank you for recognizing our efforts in resolving your last two concerns.

---

### Official Review · Reviewer_Emy3 · 2023-10-28

**Soundness:** 4 excellent
**Presentation:** 4 excellent
**Contribution:** 4 excellent
**Rating:** 10
**Confidence:** 4

**Summary:**

This paper introduces a new Model-based RL algorithm Recall to Imagine (R2I) that upgrades DreamerV3 by non-trivially incorporating S4 networks in substitutions of the RSSMs that have been commonly used since PlaNet. The paper provides abundant evidence to support the main claim that R2I outperforms DreamerV3 and is significantly more computationally efficient

**Strengths:**

While employing S4 as an alternative to RSSMs was a natural step to come, I think the authors do a very good work here. The paper is very well presented and easy to follow. Authors do a very detailed theoretical analysis of the approach, building up intuition and making it very easy to follow. It is remarkable that they even include in-depth discussions on why they discarded alternative features when designing their algorithm, which should be a more common practice. The empirical analysis is also extensive and provides solid ground with the numerous ablations and the varied array of benchmarks where the R2I vs Dreamer comparison is drawn. R2I proves to be significantly faster to train while being a better performer, specially in challenging long-term dependencies

I believe that this paper will be very relevant for the ICLR community.

**Weaknesses:**

My biggest concern is the lack of an explicit literature review/ related work section, which I would suggest to include in the appendix. Specifically, I believe that it is of special relevance to include a more in-depth comparison between R2I and S4WM -currently briefly mentioned in the conclusions- since both combine DreamerV3 with SSMs.

Also, I noticed there are no mentions about making the code available. Thus, it would be specially benefitting for reproducibility if authors include a summary of the alg in pseudocode at the appendix.

-- After Rebuttal --

Authors addressed very well all my concerns, the paper now presents clearly its differences with respect prior work and with a well documented code reproducibility should be easy to reproduce. I believe this will be a very relevant work for the RL community

**Questions:**

Correctly addressing the two points above is what can change the most my opinion. Additionally, there are a couple of minor things I noticed:
* Appendix N Figure 14, the second plot is missing the expected error
* Section 3.1 line 4 where x_t is "a" hidden state and f_θ is a sequence model with "a"  SSM
network

---

> ### Author Response · Authors · 2023-11-16
> **Author Response**
>
> Thank you for your positive and encouraging feedback on our submission. Your acknowledgment of the strong points in our paper is very uplifting, including the adept use of SSMs in MBRL, the clear and well-structured presentation, the detailed theoretical analysis, the transparency in our design choices, the extensive empirical analysis, and the performance of R2I on long-term dependencies. It is especially gratifying to know your belief that our work will be highly relevant to the ICLR community. We have reviewed the concerns expressed and are prepared to respond to each point in the following manner.
>
> ---
> > My biggest concern is the lack of an explicit literature review/ related work section.
>
> We acknowledge the need for an explicit related work section and added it to the appendix accordingly (**Section B** in the appendix). Notably, in the previous version, some literature review already exist in **Section 2** and **Appendix C**.
>
> ---
> > I believe that it is of special relevance to include a more in-depth comparison between R2I and S4WM.
>
> We appreciate your recommendation for an in-depth comparative analysis between R2I and S4WM, which is a concurrent work to ours. Recognizing the necessity to highlight the distinctiveness of R2I amidst S4WM, we acknowledge the value of such an evaluation. To address this concern, we wish to provide an analytical comparison between R2I and S4WM, demonstrating that these are significantly different algorithms. We believe S4WM makes a valuable contribution to world modeling, which is a subtask of MBRL. **Improved world modeling quality is neither necessarily nor sufficiently linked to an improved expected reward in MBRL**. To support this claim, we have added a new section (please refer to the newly added **Section Q of the Appendix**) to the Appendix, with the following discussion: First, we present both theoretical and experimental evidence from existing literature that suggests the likelihood of the world model does not directly translate to reward [1, 2, 3], and sometimes a negative correlation is observed. Second, we conduct an experiment in Memory Maze environments, which is a task of primary interest in both S4WM and R2I, where we demonstrate cases of negative correlation between the return and image MSE (i.e., the expected return improves while image MSE worsens). We also interpret how this result relates to S4WM's experimental outcomes. Third, we provide a detailed table comparing all architectural choices made by R2I and S4WM, noting there are 9 non-trivial architectural differences, excluding network sizes and depths (although these also vary). For each difference, we interpret our understanding of its impact on policy learning. Please note that our comparison is with the S4WM version that was publicly available on Arxiv on July 5th, 2023, not the Arxiv version from November 9, 2023, and not the camera-ready version from NeurIPS 2023 (whose deadline was October 27, 2023)—one month after the ICLR paper deadline. Finally, S4WM reports metrics on offline RL datasets collected by agents assumed to be nearly optimal. However, in MBRL, the world model and policy commence training with data produced by a random agent. In summary, we believe R2I and S4WM are distinct models designed to solve different (but related) tasks, released concurrently online, with no evidence that S4WM will function out-of-the-box for reinforcement learning problems—on the contrary, there is evidence suggesting that significant modifications are necessary for S4WM to operate as an MBRL algorithm.
>
> ---
> > I noticed there are no mentions about making the code available. Thus, it would be specially benefitting for reproducibility if authors include a summary of the alg in pseudocode at the appendix.
>
> Thank you for pointing out the need for enhanced reproducibility. We currently have a solid codebase, and we plan to open-source a well-documented repo along with the paper to facilitate replication of R2I. Our implementation leverages the **Jax** framework, chosen for its simplicity and efficiency. While Jax was our framework of choice, we acknowledge that the core component of our algorithm—the SSM inference with parallel scan — could potentially be implemented in pure PyTorch as well with the same efficiency as in Jax, without the need to use custom CUDA kernels, which is the case for convolution mode. Bearing this in mind and with an aim to enhance reproducibility, in the revised version, we included a detailed pseudocode for our algorithm (please refer to **Appendix S**).
>
> ---
>
> **References**
>
> [1] Joseph et al. Reinforcement learning with misspecified model classes. ICRA, 2013.
>
> [2] Lambert et al. Objective Mismatch in Model-based Reinforcement Learning. L4DC, 2020.
>
> [3] Nikishin et al. Control-oriented model-based reinforcement learning with implicit differentiation. AAAI, 2022.

---

> > ### Author Response · Authors · 2023-11-16
> > **Author Response to Questions**
> >
> > **Questions**
> >
> > ---
> > > Appendix N Figure 14, the second plot is missing the expected error.
> >
> > Thank you for your attentive review of our submission. We acknowledge the missing expected error and will include it in the next revision.
> >
> > ---
> > > Section 3.1 line 4 where x_t is "a" hidden state and f_θ is a sequence model with "a" SSM network.
> >
> > Thank you for the attentive review. The suggested corrections have been duly edited in the revised version of the paper.
> >
> > ---
> > **In conclusion, we truly value your reviews. We hope that the revisions and clarifications will influence and improve your overall opinion. If we've managed to resolve your principal concerns and questions, we'd be thankful for your endorsement through an elevated score of our submission. On the other hand, if there are remaining issues or questions on your mind, we're more than willing to address them.**

---

> > > ### Comment · Reviewer_Emy3 · 2023-11-20
> > > **Rebuttal response**
> > >
> > > I want to thank the authors for putting so much work in the response of our comments and addressing my concerns.
> > >
> > > Specifically, regarding S4WM, since it was a concurrent work with similar "ideas" but towards a different objective within model-based RL, I was simply asking for a longer discussion (in the fashion to the response to my comment), since the previous brief comment in the last section was leaving the reader puzzled about the differences. Nevertheless, the authors went a few extra miles with a detailed and in-depth theoretical and empirical analysis included in the extended version, which is very insightful. I believe this plus the additional related work section helps to picture better the role of this work.
> > >
> > > I am raising my score since, as an RL researcher, this is the kind of work I wouldn't want to miss when attending ICLR, and that I am very happy to have had the chance to review. Congratulations to the authors, I am keen to see that code released and start using R2I.

---

> > > > ### Author Response · Authors · 2023-11-21
> > > > **Thank you!**
> > > >
> > > > Thank you for recognizing the work we have put into addressing the concerns you raised. We are thrilled that the additional analysis and extended version of our paper met your standards, and that you found it insightful. Your positive feedback and the improved score are deeply appreciated. Also, your input was helpful in refining our paper. Rest assured, the code will be released, and we hope it will contribute positively to the RL community.

---

> > > > > ### Comment · Reviewer_Emy3 · 2023-11-22
> > > > > **Small thing**
> > > > >
> > > > > Just a small typo I just saw, equation 5 one of the z is missing a hat since should be the KL diff between the encoded and the predicted z as correctly reflected in Figure 1

---

> > > > > > ### Author Response · Authors · 2023-11-22
> > > > > > **Author Response**
> > > > > >
> > > > > > Thank you for noticing the typo in the main objective of the model. We will update the paper to fix this typo (in the camera-ready version of the paper, in case it gets accepted).

---

### Author Response · Authors · 2023-11-15
**Official response to all reviewers**

We gratefully acknowledge all the reviewers for their insightful comments. We appreciate the recognition from reviewers of clear and good presentation (**Reviewers Emy3, pPqM, fFhp**), improved performance in RL memory-intensive tasks (**Reviewers Emy3, pPqM, fFhp**), state-of-the-art results (**Reviewer pPqM**), extensive experiments and ablations (**Reviewers Emy3, pPqM, fFhp**), as well as the rich and detailed technical analysis of the presented method (**Reviewers Emy3, pPqM**). We also wish to distinctly thank **Reviewer Emy3** for the recognition of the in-depth discussion of design choices and the rationale for discarding alternative features, which **Reviewer Emy3** noted as a commendable practice deserving to be used more commonly.

As observed by all reviewers (**Reviewers Emy3, pPqM, fFhp**), there is a related work, specifically S4WM, which was published online on July 5, 2023. This study solves a different—yet relevant—task of world modeling. However, we wish to emphasize that according to ICLR's public review guidelines, any papers released online within 4 months of the submission deadline are regarded as contemporaneous, and authors are not mandated to compare their work with such papers. S4WM falls into this category.

Despite this guideline, we have included a theoretical analysis of the interrelations between R2I and S4WM in our revised submission (detailed in **Appendix Q**). Specifically, the analysis includes the following:

1. An examination of insights from existing literature, which indicates that enhanced world modeling (a primary task of interest in S4WM) does not necessarily translate to improved model-based RL performance.
2. The design and results of a synthetic experiment where the main metric of interest of S4WM (image mean-squared-error) worsens, while the primary metric for RL methods and R2I (expected return) shows improvement.
3. An explicit distinction between R2I and S4WM by presenting a list of 9 significant algorithmic distinctions between the two and explaining how each of them can facilitate improved reinforcement learning.

The latest revision of our paper has been uploaded, addressing all comments and queries raised by the reviewers. Additionally, we have added a sole experimental update in which we further trained R2I over an extended memory length in the Memory Maze. We would like to clarify that the algorithm was not changed (compared to our initial submission), while **R2I now establishes the new absolute state-of-the-art in the whole Memory Maze domain, surpassing all prior baselines across all difficulty levels**.

In summary, the revised paper incorporates the following updates, all of which are highlighted in blue for the reviewers' ease of reference:

1. Addition of a discrete related work section (**Appendix B**) in response to **Reviewer Emy3's** feedback.
2. Inclusion of a new section (**Appendix Q**) that delves into the correlation between the likelihood of the world model and agent performance, alongside a comprehensive comparison of R2I and S4WM, as per the discussion.
3. Inclusion of a new section (**Appendix R**) providing a detailed and reliable comparison of R2I and DreamerV3 across standard reinforcement learning benchmarks, directly addressing concerns raised by Reviewers pPqM and fFhp regarding the generality of the results.
4. Integration of a new section (**Appendix S**) depicting the pseudocode details, fulfilling a request from **Reviewer Emy3**.
5. Expansion of the content with a new section (**Appendix T**) demonstrating the SSM-based model-free RL approach employed in the POPGym environment, as inquired by **Reviewer pPqM**.
6. Revision of figures: Inclusion of the “PPO+S4D” baseline in **Figure 4** to address a suggestion from **Reviewer pPqM**, and an update to **Figure 5** showcasing the enhanced results of R2I.
7. Minor textual modifications throughout the paper, such as an explicit mention of Dreamer's objective (per **Reviewer fFhp’s** request), a clear assertion that Dreamer employs truncated backpropagation through time (TBTT) in the Memory Maze experiments, and a more accurate statement about the generality in **Section 4.3**, in response to the comments from **Reviewers pPqM and fFhp**.

These revisions have been made to address the valuable feedback provided by the reviewers and to enhance the clarity, accuracy, and depth of the paper.

---

### Meta-Review · Area_Chair_bqCR · 2023-12-02

**Metareview:**

This paper contributes a new RL algorithm which achieves state of the art performance on some standard benchmarks. The improvement over standard approaches like IMPALA seems fairly large. It also improves upon the newer Dreamer family of algorithms. All reviewers were in agreement that this paper should be accepted.

**Justification For Why Not Higher Score:**

NA

**Justification For Why Not Lower Score:**

The only negatives raised about the paper were mainly in connection to novelty, but they seemed to be smoothed over in the discussion phase. All the reviewers seemed satisfied with the author's responses.

---

### Decision · Program_Chairs · 2024-01-16

Accept (oral)